# A connectional hub in the rostral anterior cingulate cortex links areas of emotion and cognitive control

Wei Tang[1], Saad Jbabdi[2], Ziyi Zhu[3], Michiel Cottaar[2], Giorgia Grisot[4], Julia F Lehman[3], Anastasia Yendiki[4], Suzanne N Haber[1,3]*

[1]McLean Hospital, Harvard Medical School, Belmont, United States; [2]Centre for Functional MRI of the Brain, Department of Clinical Neurology, University of Oxford, Oxford, United Kingdom; [3]Department of Pharmacology and Physiology, University of Rochester School of Medicine & Dentistry, Rochester, United States; [4]Athinoula A Martinos Center for Biomedical Imaging, Massachusetts General Hospital, Harvard Medical School, Charlestown, United States

**Abstract** We investigated afferent inputs from all areas in the frontal cortex (FC) to different subregions in the rostral anterior cingulate cortex (rACC). Using retrograde tracing in macaque monkeys, we quantified projection strength by counting retrogradely labeled cells in each FC area. The projection from different FC regions varied across injection sites in strength, following different spatial patterns. Importantly, a site at the rostral end of the cingulate sulcus stood out as having strong inputs from many areas in diverse FC regions. Moreover, it was at the integrative conjunction of three projection trends across sites. This site marks a connectional hub inside the rACC that integrates FC inputs across functional modalities. Tractography with monkey diffusion magnetic resonance imaging (dMRI) located a similar hub region comparable to the tracing result. Applying the same tractography method to human dMRI data, we demonstrated that a similar hub can be located in the human rACC.

DOI: https://doi.org/10.7554/eLife.43761.001

*For correspondence:
Suzanne_Haber@urmc.rochester.edu

**Competing interests:** The authors declare that no competing interests exist.

## Introduction

The anterior cingulate cortex (ACC) is composed of multiple regions that support a wide range of functions (emotion, motivation, higher cognition, and motor control), and thus, is in a position to use value-related information to help regulate flexibility, adaptation and top-down control (*Etkin et al., 2015*; *Kolling et al., 2016*; *Shenhav et al., 2016*). This functionally heterogeneous region is anatomically divided into the subgenual ACC (sACC), the rostral ACC (rACC), and the dorsal ACC (dACC) (*Morecraft et al., 2012*; *Morecraft and Tanji, 2009*; *Ongür and Price, 2000*). The sACC is connected to the motivation network consisting of the orbitofrontal cortex (OFC) and the amygdala. It is involved in visceral and emotional functions, an important mediator of motivation, and is critical for determining value (*Camille et al., 2011*; *Jocham et al., 2012*; *Kolling et al., 2016*). The rACC is tightly linked with both the sACC and dACC, the dorsolateral and ventrolateral prefrontal cortex (dlPFC and vlPFC), and is associated with cognitive control and choice of action (*Jiang et al., 2015*; *Kolling et al., 2018*). Caudally, the dACC is connected with the action network consisting of motor control areas, including frontal eye fields (FEF) and premotor areas (*Morecraft et al., 2012*; *Ongür and Price, 2000*). The dACC is associated with motor planning and action execution (*Caruana et al., 2018*; *Picard and Strick, 1996*). Importantly, there are no clearly defined borders between the sACC, rACC and dACC based on their anatomical connections. Regions that project

strongly to an ACC subdivision also projects to the others, albeit to a lesser degree (*Morecraft and Tanji, 2009*).

The rACC sits at the connectional intersection of the motivation and action control networks. Thus, it is in an important position in the transition from valuation, to choice, to action. An important question is how the transitions occur within the rACC. One possibility is that information processing changes sequentially across subregions within the rACC, from valuation in regions close to the sACC, to cognition at the center of the rACC, and then to action in regions close to the dACC (*Kable and Glimcher, 2009*; *Rangel and Hare, 2010*; *Shadlen et al., 2008*). Alternatively, different functional processing may be integrated in a central location, or hub (*Cisek, 2012*; *Hunt et al., 2013*; *Kolling et al., 2012*; *Lee et al., 2014*; *Rushworth et al., 2012*). In network theory, hubs are nodes of a network that have unusually high connectivity to other nodes (a.k.a. degree-centrality) and high connectivity to other hubs (a.k.a. eigenvalue-centrality). In the context of cortical networks, the connectivity that defines a hub is not simply the intersection of shared inputs from functionally similar or adjacent cortical regions. Rather, the connectivity of a hub is higher and more diverse than its neighbors. Hubs thus represent regions for integrating and distributing information from multiple regions (*van den Heuvel and Sporns, 2013*). In this regard, the entire rACC can be considered a hub (*Buckner et al., 2009*). However, as the rACC is a large area, we sought to determine whether there is a specific region within it that is uniquely positioned to integrate signals across several functional domains. Networks in this study are based on anatomical connections, which allows for measurement of degree-centrality but not eigenvalue-centrality. We thus define the hub according to its high degree of inputs and its position in the network for cross-domain integration.

To determine whether a hub region exists in the rACC, we systematically placed tracer injections along the rACC in nonhuman primates (NHP), and quantified the input strength and patterns from the frontal cortex (FC) to each injection site. As expected, projections from the vmPFC were concentrated in the ventral rACC and those from motor control areas were found more dorsally and caudally. Projections from the vlPFC, dlPFC, dmPFC and OFC varied across sites. However, one site showed uniquely high degree-centrality with the FC, suggesting a hub region at this site. We verified that inputs to this site were from functionally diverse regions and were not a composition of inputs to the neighboring cytoarchitectonic areas. To test whether a similar connectivity pattern and the location of a hub could be identified in the human rACC, we used dMRI tractography to analyze connections between the FC and the rACC. Comparing two connectivity modalities (tracing and dMRI) in different species (NHP and humans) is problematic. We therefore first verified that dMRI correctly detected connectivity convergence in NHPs. Note: for two experiments, the tracer injections and dMRI were carried out in the same animals. We seeded each frontal area and, consistent with the tracing results, probabilistic streamlines converged in a similar location as the hub within the rACC in the NHP. We then seeded each frontal area in a dMRI dataset from the Human Connectome Project (HCP). As in the NHP results, the streamlines converged in a similar rACC location.

## Materials and methods

### Overview

Bidirectional tracer injections were placed systematically throughout the rACC. FC was divided based on cytoarchitectonics into areas (*Pandya and Seltzer, 1982*; *Paxinos et al., 2000*; *Preuss and Goldman-Rakic, 1991*; *Vogt, 2009b*; *Vogt, 1993a*) and retrogradely labeled cells were quantified using StereoInvestigator. To normalize for comparison across cases (variability in uptake and transport), we calculated the percentage of total labeled cells that projected from each area to each injection site. These percent scores were independent from the size of each area (Figure 7—figure supplement 1A). The areas were further pooled into major FC regions according to their associated functions. Using the percent scores as the measurement for input strength, projection gradient across injection sites was analyzed for each FC region. Guided by the tracing results, probabilistic tractography was conducted on the NHP and human dMRI. Cortical areas were used as seed masks and the rACC as the target mask. The connectivity strength was determined by the number of streamlines between each target voxel and an FC area, divided by the total number of streamlines from that area to all rACC voxels. A convergent-connectivity value was calculated for each rACC voxel as the sum of connectivity strength weighted by the number of areas with non-zero

streamlines. We identified voxels with the highest convergent-connectivity value for each individual subject and compared their locations to the hub region found in tract tracing.

## Anatomical tracing experiments

All experiments were performed in accordance with the Institute of Laboratory Animal Resources Guide for the Care and Use of Laboratory Animals and approved by the University Committee on Animal Resources at University of Rochester. Animals were adult male monkeys (*Macaca mulatta* and *Macaca fascicularis*). Bidirectional tracers were injected into the rACC. Details of the surgical and histological procedures have been described previously (*Haber et al., 2006*; *Safadi et al., 2018*). Monkeys were first tranquilized by intramuscular injection of ketamine (10 mg/kg) and then maintained anesthetized via 1–3% isoflurane in 100% oxygen. Temperature, heart rate, and respiration were monitored throughout the surgery. Pre-surgery structural MR images were used to locate the stereotaxic coordinates for the injection sites. Monkeys were placed in a David Kopf Instruments stereotax, a craniotomy (2–3 cm) was made over the region of interest, and small dural incisions were made at injection sites. Bidirectional tracers (40–50 µl, 10% in 0.1 mol phosphate buffer (PB), pH 7.4; Invitrogen) were pressure injected over 10 min using a 0.5 µl Hamilton syringe, separate for each case. Tracers used for the present study were Lucifer Yellow (LY), Fluororuby (FR), or Fluorescein (FS). The tracers were all conjugated to dextran amine (Invitrogen) and had similar transport properties (*Rajakumar et al., 1993*). After each injection, the syringe remained in situ for 20–30 min.

After a survival period of 12–14 days, monkeys were again deeply anesthetized and perfused with saline, followed by a 4% paraformaldehyde/1.5% sucrose solution in 0.1 mol PB, pH 7.4. Brains were postfixed overnight and cryoprotected in increasing gradients of sucrose (10, 20, and 30%; *Haber et al., 2006*). Brains were removed and shipped to the Martinos Center for Biomedical Imaging. Diffusion MRI data was collected with the brains submerged in Fomblin solution to eliminate susceptibility artifacts at air-tissue interfaces and background signal (see *dMRI data collection and analysis* for imaging protocols). After imaging, the brains were shipped back to University of Rochester Medical Center for histological processing. Serial sections of 50 µm were cut on a freezing microtome, and one series with sections 1.2 mm-apart was processed for subsequent retrograde tracing. The serial sections were processed free-floating for immunocytochemistry. Tissue was incubated in primary anti-LY (1:3000 dilution; Invitrogen), anti-FS (1:1000; Invitrogen), or anti-FR (1:1000; Invitrogen) in 10% NGS and 0.3% Triton X-100 (Sigma-Aldrich) in PB for 4 nights at 4°C. After extensive rinsing, the tissue was incubated in biotinylated secondary antibody followed by incubation with the avidin-biotin complex solution (Vectastain ABC kit, Vector Laboratories). Immunoreactivity was visualized using standard DAB procedures. Staining was intensified by incubating the tissue for 5–15 s in a solution of 0.05% DAB tetrahydrochloride, 0.025% cobalt chloride, 0.02% nickel ammonium sulfate, and 0.01% $H_2O_2$. Sections were mounted onto gel-coated slides, dehydrated, defatted in xylene, and coverslipped with Permount. In cases in which more than one tracer was injected into a single animal, adjacent sections were processed for each antibody reaction.

## Analysis

*strength of inputs and defining the hub.* Nine out of sixteen injection sites were selected for analysis based on the following criteria (*Figure 1*): 1. location of the injection site along the rACC; 2. lack of tracers leaking into adjacent cortical regions or into the white matter; 3. outstanding retrograde transport; and 4. low background. We evaluated the transport by known and distant connections of the rACC, including the posterior cingulate/retrosplenial cortex and thalamus. Locations 2, 5 and 6 (*Figure 1A*) had overlapping injection sites. Cell labeling patterns were surveyed to ensure similarity between the overlapping sites, and one site from each location (six in total, see *Figure 1A*) was chosen for analysis. In addition, three injection sites close to site 5 were used as controls: 1. a large injection controlled for size; 2. a dorsal bank injection controlled for dorsoventral variability; 3. an injection in the left hemisphere controlled for the variability across studies. To evaluate the strength of the different FC inputs to each site, we divided the FC into 27 areas based on the atlas by *Paxinos et al. (2000)*, in conjunction with detailed anatomical descriptions (*Table 1*) (*Pandya and Seltzer, 1982*; *Preuss and Goldman-Rakic, 1991*; *Vogt, 2009c*; *Vogt, 1993b*). Labeled cells were quantified throughout the FC using StereoInvestigator software (MicroBrightField) as previously described (*Choi et al., 2017*). For each case, the stack of 2D coronal charts from StereoInvestigator

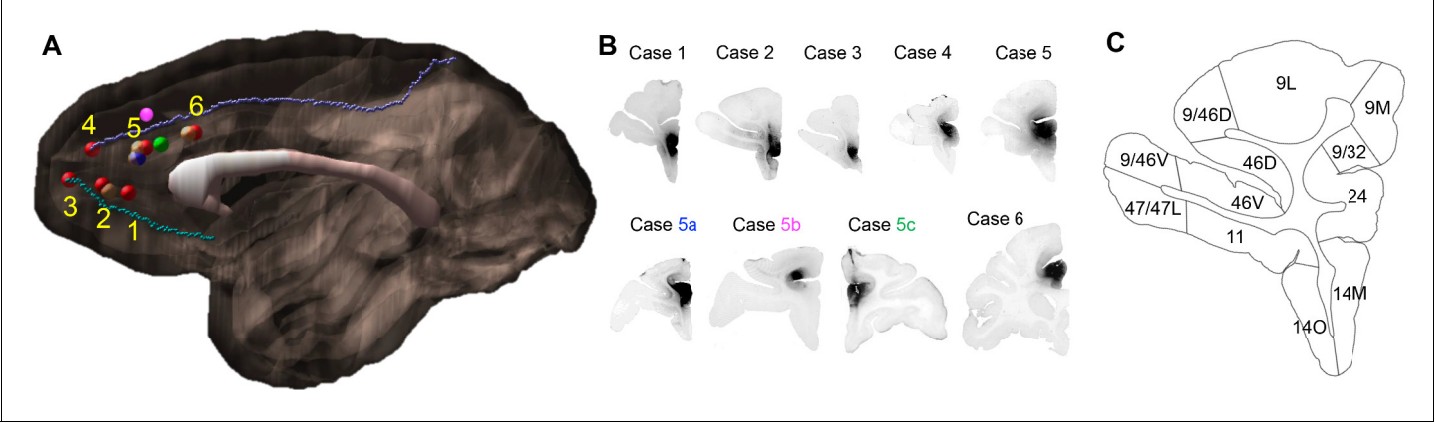

**Figure 1.** Injection sites and cortical area definition. (**A**) Locations of the 9 out of 16 rACC injections that were selected for stereology analysis. Six cases (red) were analyzed as the main result and the additional three (blue = 5a, pink = 5b, green = 5c) as control. The numbering of cases followed the longitudinal axis of the cingulate gyrus. Cases not used for stereology due to the overlap with sites at the same location were colored beige. The cortex of the right hemisphere is shown in semi-transparent brown, the corpus callosum in white, and the cingulate sulcus and rostral sulcus in purple and cyan. (**B**) Photomicrographs to show the histology of each injection site. (**C**) Section outline of FC areas on an example slide. The borders between areas were hand-drawn following the atlas by *Paxinos et al. (2000)*.

DOI: https://doi.org/10.7554/eLife.43761.002

were imported into IMOD, a 3D rendering program (Boulder laboratory for 3D Electron Microscopy). A 3D reconstruction of each cell was created for each case separately. The 3D model of each case was then merged through spatial alignment to our in-house template brain as previously described (*Haber et al., 2006*).

To compare the input pattern across injection sites, the percent input to each site was calculated based on the number of labeled cells in each FC area projecting to a given site, divided by the total number of labeled cells across all FC areas projecting to the same site. Areas were then ordered based on their percent scores. The number of areas whose cell counts added up to 50% and 75% of total input was calculated for each site. To determine whether the inputs were distributed evenly across areas or highly concentrated in a few areas, an entropy score was calculated to compare the input pattern to a uniform distribution:

$$H = -\sum_{i=1}^{27} c_i \log(c_i) \tag{1}$$

where $c_i$ is the percent cell count for the *i*-th area, and log() is the natural logarithm function (*Conrad, 2004*). The hub was characterized by a high number of areas contributing to 50% and 75% of total inputs. Additionally, the hub was expected to have a high entropy score that indicates evenly distributed inputs across areas.

## FC regions

FC areas were grouped into commonly defined regions: frontal pole (FP, areas 10M, 10L), vmPFC (areas 14M, 25), OFC (areas 14O, 11, 13, OPAl, OPro), vlPFC (areas 47L, 47O, 44, 45A, 45B), dlPFC (areas 9L, 46, 9/46V, 9/46D), dmPFC (areas 9/32, 9M), frontal eye fields (FEF, areas 8/32, 8A, 8B) and premotor cortex (areas 6M, 6L, ProM, 6/32) (see e.g. *Clark et al., 2010*). The demarcation and definition of vmPFC vary across studies and can include the medial OFC, medial and ventral part of area 10, and area 25. Moreover, area 10 also contains a dorsal and a lateral component, which together with the medial and ventral parts cover the entire polar region. To maintain cytoarchitectonic consistency in spatial demarcation, we separated area 10 (designated as FP) from vmPFC.

**Table 1.** List of the 27 frontal areas.

| FP | vmPFC | dmPFC | OFC | vlPFC | dlPFC | FEF | Premotor |
|----|-------|-------|-----|-------|-------|-----|----------|
| 10L | 14M | 9M | 14O | 47L | 9/46V | 8A | 6L |
| 10M | 25 | 9/32 | 11 | 47O | 9/46D | 8B | 6M |
| | | | 13 | 44 | 46 | 8/32 | 6/32 |
| | | | OPro | 45A | 9L | | ProM |
| | | | OPAl | 45B | | | |

DOI: https://doi.org/10.7554/eLife.43761.003

## Comparison of the inputs to site 4 with the composition of inputs to areas 32 and 24

Based on the analyses described above, site 4 stood out as having a uniquely diverse input pattern as compared to the other sites. Such connectivity property suggested that site 4 is a hub in the FC-rACC network. However, because site 4 was at the junction of areas 32 and 24, an important question is whether its input pattern could be explained by a composition of inputs to areas 32 and 24. We tested this alternative hypothesis by comparing the connectivity pattern of site 4four with mixtures of the patterns of areas 32 and 24 using the symmetric Kullback-Leibler distance. The percent cell counts for sites 1, 2 and 3 and those for sites 5 and 6 were averaged to represent the input distributions of areas 32 and 24, respectively. The composition of the two input distributions followed a weighted sum:

$$P_{s4}(a) = wP_{32}(a) + (1 - w)P_{24}(a) \tag{2}$$

where $P_{32}(a)$ and $P_{24}(a)$ are the probabilities of observing a labeled cell in area $a$ according to the input distributions of areas 32 and 24, respectively. $P_{s4}(a)$ is the probability composed of $P_{32}(a)$ and $P_{24}(a)$ with weight $w$ ($0 \leq w \leq 1$). When $w$ is held constant across areas, *Equation (2)* provides a linear combination of two input distributions. We also tested non-linear mixing by assigning a different $w$ to each area. The values of $w$ were randomly sampled from a uniform distribution over the [0, 1] interval.

## dMRI data collection and analysis

The dMRI data collection and preprocessing was previously described in *Safadi et al. (2018)*. The NHP dMRI data was collected from seven animals in a small-bore 4.7T Bruker BioSpin MRI system, with gradient internal diameter of 116 mm, maximum gradient strength 480 mT/m, and birdcage volume RF coil internal diameter of 72 mm. Two of the animals also had tracer injections (cases 1 and 6). A 3D echo-planar imaging (EPI) sequence was used for dMRI with TR = 750 ms, TE = 43 ms, $\delta$ = 15 ms, $\Delta$ = 19 ms, $b_{max}$ = 40,000 s/mm$^2$, 514 gradient directions, matrix size 96 $\times$ 96 $\times$ 112, and 0.7 mm isotropic resolution. The human dMRI data used 35 healthy subjects, publicly available as part of the MGH-USC Human Connectome Project (HCP) (*Fan et al., 2016*). Both the NHP and human data were preprocessed using FSL 5.0.9 (*Jenkinson et al., 2012*). Artifacts of head movements and distortions by eddy currents were corrected (*Andersson and Sotiropoulos, 2016*). A crossing fiber model (bedpostx) (*Behrens et al., 2007*) was fit to each voxel to estimate the distribution of fiber orientations, which were subsequently used in the probabilistic tractography.

Following preprocessing, probabilistic tractography (*Behrens et al., 2007*) was performed in each individual's diffusion space, and the results were transformed to a template for comparison across subjects. Parcellation of the FC areas for NHP followed the same atlas used for tract tracing (*Paxinos et al., 2000*). Each FC area was used as a seed, while ACC areas 32 and 24 were combined and used as the target. Twenty-seven seed masks were created correspondent to the 27 FC areas used in the tracing analysis. To generate seed and target masks, areal masks from a template brain (*Calabrese et al., 2015*) were transformed to each individual's diffusion space via nonlinear registration (*Klein et al., 2010*). Each mask covers 0.14 mm thickness (two voxels) of white matter at the gray-white matter boundary (Figure 7B). Anatomical parcellation of the human cortex was hand drawn on a surface-based template (*fsaverage* by FreeSurfer 4.5, *Supplementary file 1*). The parcellation followed *Petrides et al. (2012)*, which was developed to maximize architectonic

correspondence between human and NHP prefrontal areas (see Discussion on the cross-species homologies). Twenty-five hand-drawn masks were transformed from the *fsaverage* space to each individual's diffusion space with nonlinear registration provided by FreeSurfer. The FC areas were used as seeds, and areas 32 and 24 were combined and used as the target. Each mask covers 3 mm thickness (two voxels) of white matter at the gray-white matter boundary. The *fsaverage* parcellation is available in *Supplementary file 1*.

FSL was used for probabilistic tractography for both NHP and human. Tractography was performed from each seed mask to the ipsilateral target mask. At each voxel, a sample was drawn from the orientation distribution of the anisotropic compartment with the closest orientation to the previously visited voxel (*Behrens et al., 2007*). To exclude indirect pathways through subcortical structures, the thalamus, the striatum and the amygdala were used as exclusion masks. To measure the connectivity strength between each seed mask and each target voxel, the number of streamlines arriving at the target voxel was divided by the total number of streamlines generated from the seed mask. The convergent-connectivity value was defined as the sum of areas with non-zero streamlines multiplied by the connectivity strength. The final results from all individuals were displayed on a template brain via linear registration (FA image of the template by *Calabrese et al., 2015* for NHP, T1-weighted image of the MNI152 template for human).

## Results

### Overview

Injection sites were labeled 1–6 based on the spatial order of their locations (*Figure 1*). We found that the labeling of cortical areas in cases that overlapped with sites 2, 5 and 6 were consistent with these three sites. Specifically, we did not observe FC areas that contained labeled cells in the non-selected cases but not in the selected cases, or the reverse. Sites 1–3 were in area 32 and ordered from caudal to rostral (FR, LY and FS, respectively). Site 4 (FR) was in the transition zone between areas 32 and 24. Sites 5 (LY) and 6 (FR) were in area 24. There were three control cases, 5a (FR), 5b (FS) and 5c (LY). Case 5a was particularly large, located in area 24, and overlapped extensively with site 5. This case served as a control for the variability of injection size. Injection 5b was in the dorsal bank of the cingulate sulcus above area 24. It was analyzed to determine the variability of inputs to the dorsal bank of the cingulate sulcus. Injection 5c was in the left hemisphere caudal to site 5. This case matched most closely with a case previously reported in the literature and therefore was used to compare our general findings with previously reported data. All cases showed dense labeling in areas 23, 29 and 30, and the AV, VA and MD nucleus in the thalamus. Such labeling demonstrated efficient transport. Following quantification of the cell labeling into percent counts, we found that the sites differed with respect to projection strength from various FC areas. The results demonstrated three projection patterns across sites: 1. a decrease in the input strength to sites 1–6 from FP and vmPFC; 2. an increase in the input strength to sites 1–6 from FEF and premotor cortex; and 3. a non-monotonic gradient centered on site 4 with the input strength from dlPFC, dmPFC and vlPFC increasing to sites 1–4 and decreasing to sites 4–6. Taken together these data showed that site 4 received inputs from the greatest number of FC areas. Site 4 also contained the most regionally diverse inputs compared to the other sites. Finally, consistent with the tracing results, dMRI tractography in NHP and human showed convergent probabilistic tracts from the FC to an rACC region in close proximity to site 4.

### Projection patterns from the FC areas to the rACC

Not all FC regions project equally to the rACC. Overall, retrogradely labeled cells in cases 1–3 were primarily located in more rostral regions of the FC, while those in cases 4–6 were primarily in more caudal regions (*Figure 2*). Case 1 showed high concentration of labeled cells at the rostral pole of the medial PFC. However, at more caudal levels, the concentration of labeled cells was located primarily ventrally, with few labeled cells in areas 9L and 9/32. In addition, there were clusters of labeled cells located in a caudal OFC area OPro. Similarly, in cases 2 and 3, labeled cells were concentrated rostrally in FP and vmPFC regions. However, some clusters with a few labeled cells were also located more dorsally in area 9L and caudally in the lateral OFC. In contrast to case 1, additional labeled cells were found in the vlPFC (areas 47L and 47O). Case 4 showed a diverse pattern to the

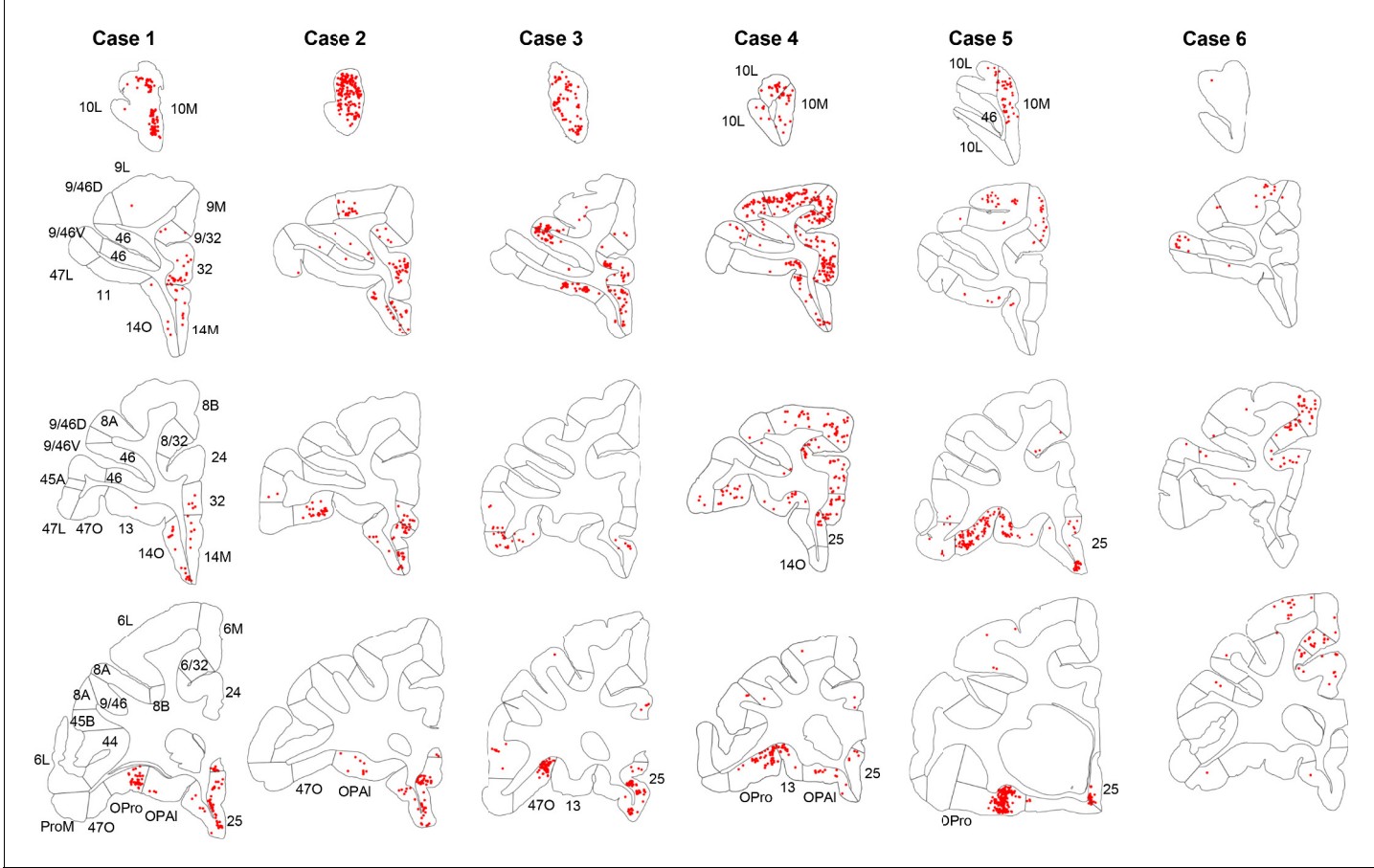

**Figure 2.** Retrogradely labeled cells (red) in cases 1–6.  Each column contains rostral to caudal coronal sections from one case. Sections of the same row have matching locations along the rostro-caudal axis. FC areas are labeled in the first column. Areas not matching the parcellation in the first column were labeled additionally on the corresponding sections.

DOI: https://doi.org/10.7554/eLife.43761.004

distribution of labeled cells, with a clear increase of dense clusters of labeled cells located in dorsal regions. At the most rostral level, labeled cells were concentrated in FP. At more caudal levels, labeled cells were extensively distributed in dlPFC, vlPFC, ventromedial OFC and vmPFC. An additional cluster of labeled cells was also found in area OPro. Case 5 resulted in labeled cells concentrated ventrally, primarily in the caudal part of orbital and medial regions, although scattered clusters of labeled cells were also located rostrally in the FP and dorsally in area 9L. Finally, labeled cells in case 6 were primarily distributed in caudal dorsal regions, in area 8 and premotor cortex. Only very few labeled cells were found in ventral or rostral regions.

An essential property of network hubs is their high degree of connections (degree-centrality). To measure the degree-centrality of each site, we identified the number of areas that contributed to 50% and 75% of total inputs. This number varied extensively across sites (*Figure 3*). At site 1, only two areas, 25 and 10M accounted for 50% of the total inputs. Inputs from areas 10L and OPro accounted for an additional 25%, for a total of four areas comprising 75% of all inputs to site 1. Similarly, two areas, 10L and 10M contributed 50% of total inputs to site 2, and two areas 25 and 46 accounted for an additional 25%, for a total of four areas comprising 75% of all inputs to site 2. At site 3, three areas 25, 46 and 47O contributed 50% of the total inputs, with three additional areas 9L, 10L and 11 (the additional 25%) for a total of six areas comprising 75% of all inputs to site 3. In contrast to sites 1–3, site 4 received 50% of the total inputs from five FC areas 9L, 47O, 9M, OPro and 46. Five areas, 9/32, 14O, 14M, 8B and 9/46D, contributed the additional 25%, for a total of ten areas comprising 75% of all inputs to site 4. More consistent with sites 1 and 2, site 5 received 50%

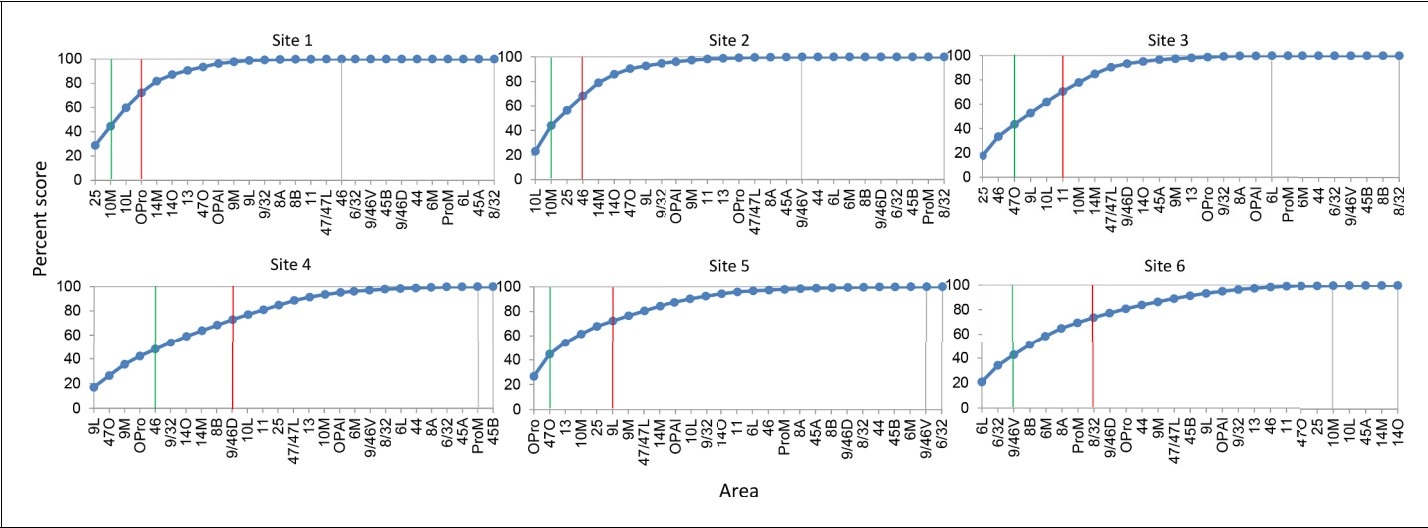

**Figure 3.** Cumulative percent cell count across areas in cases 1–6. Cutoff remarks: green line = 50%, red line = 75%, gray line = 100%. Areas after 100% are in a random order. The slope of each curve indicates deviation from a uniform distribution (steeper slope = further deviation).
DOI: https://doi.org/10.7554/eLife.43761.005

of its inputs from only two areas, OPro and 47O. Four areas, 13, 10M, 25 and 9L contributed the remaining 25%, for a total of six areas comprising 75% of all inputs to site 5. Finally, three areas, 6L, 6/32 and 9/46V, contributed 50% of total inputs to site 6. Five areas, 8B, 6M, 8A, ProM and 8/32 contributed an additional 25%, for a total of eight areas comprising 75% of all inputs to site 6. In summary, the number of areas accounting for 50% and 75% of total inputs varied across sites (*Table 2*). The curves generated by calculating the number of areas contributing 50% of the total inputs to each site illustrated the relatively limited number of inputs to sites 1–3 and 5 compared to sites 4 and 6 (*Figure 3 and 4A*). The entropy measure of cell counts across areas was to verify that the difference of cell distribution across sites was not due to an arbitrary percentage cutoff. Consistent with the area counts in *Figure 4A*, the entropy score was the lowest at sites 1 and 2, and the highest at site 4 and site 6 (*Figure 4B*).

## Projection patterns from cortical regions to the rACC

FC areas were grouped into commonly referred FC regions. These included: FP (areas 10M, 10L), vmPFC (areas 14M, 25), OFC (areas 14O, 11, 13, OPAl, OPro), vlPFC (areas 47L, 47O, 44, 45), dlPFC (areas 9L, 46, 9/46), dmPFC (areas 9/32, 9M), FEF (areas 8/32, 8A, 8B), and premotor cortex (areas 6, ProM, 6/32). At each site, we identified the FC regions contributing 50% and 75% inputs (*Figure 5*). FP and vmPFC contributed 50% of the total inputs to site 1. The next 25% were from FP and OFC, for a total of 3 regions. At site 2, only FP contributed 50% of the inputs. Inputs from vmPFC and dlPFC comprised the next 25% for a total of 3 regions. vmPFC, vlPFC and dlPFC contributed

**Table 2.** Summary of the major FC inputs to each site.

| Site | # areas contributing 50% inputs | Regions contributing 50% inputs | # areas contributing 75% inputs | Regions contributing additional 25% inputs |
|---|---|---|---|---|
| 1 | 2 | FP, vmPFC | 4 | FP, OFC |
| 2 | 2 | FP | 4 | vmPFC, dlPFC |
| 3 | 3 | vmPFC, vlPFC, dlPFC | 6 | FP, OFC, dlPFC |
| 4 | 5 | OFC, vlPFC, dmPFC, dlPFC | 10 | vmPFC, OFC, dmPFC, dlPFC, FEF |
| 5 | 2 | OFC, vlPFC | 6 | FP, vmPFC, OFC, dlPFC |
| 6 | 3 | dlPFC, FEF, Premotor | 8 | FEF, Premotor |

DOI: https://doi.org/10.7554/eLife.43761.007

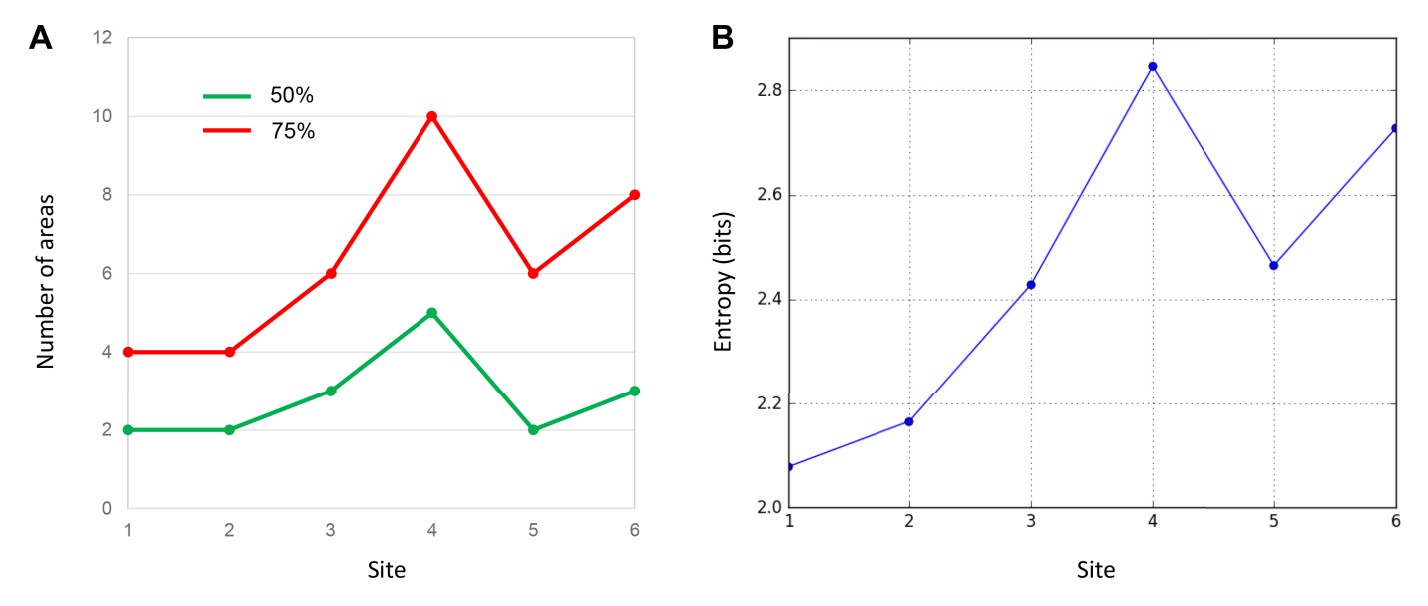

**Figure 4.** Quantitative comparison of cell distributions in cases 1–6. (**A**) The number of areas that contributed 50% (green) and 75% (red) of inputs in each case. The highest number was found in case 4. (**B**) The entropy of cell distribution in each case. Higher entropy corresponds to less deviation from the uniform distribution, that is more evenly distributed labeled cells across areas. The highest entropy was in case 4.
DOI: https://doi.org/10.7554/eLife.43761.006

50% inputs to site 3, and OFC and FP the next 25% for a total of 5 regions. OFC, vlPFC, dmPFC and dlPFC contributed 50% of the total inputs to site 4. The next 25% were from vmPFC, OFC, dmPFC, dlPFC and FEF, for a total of 6 regions. Inputs from OFC and vlPFC comprised 50% of the inputs to sites 5. The next 25% were from FP, vmPFC, OFC, and dlPFC, for a total of 5 regions. Finally, at site 6, dlPFC and premotor cortex contributed 50% of the inputs. The next 25% were from FEF and premotor cortex for a total of 3 regions. In summary, sites 1, 2 and 5 had the most limited regional input, and site 4 had the most diverse regional input.

## Projection trends from FC regions to the rACC

Using the mean percent score for the projection strength from each FC region to each site, we identified three modes of cortical projection patterns to the rACC (*Figure 6*). There were two monotonic trends across all sites, one related to the FP and vmPFC projections, the other the premotor and FEF projections. The third mode was a nonmonotonic gradient with a single peak at site 4, related to the dlPFC, vlPFC, and dmPFC projections. In the first trend, vmPFC and FP contributed to more than 15% of inputs to sites 1 and 2,~10% to site 3, just under 5% to sites 4 and 5, and close to 0% to site 6 (*Figure 6A*). The between-site difference was statistically significant (Kruskal-Wallis $H = 10.48$, p<0.02). This trend indicated a decrease in projection strength of vmPFC and FP along a gradient from site 1 to 6. In contrast, the second trend demonstrated an increase of projection strength from site 1 to 6 of inputs from premotor cortex and FEF. These regions projected strongly to site 6, but weakly to the other sites. Indeed, the average contribution of an area in these regions was less than 1% to the inputs to sites 1–3,~2% to sites 4 and 5, and more than 5% to site 6 (*Figure 6B*). The between-site difference was also statistically significant (Kruskal-Wallis $H = 35.46$, $p < 1 \times 10^{-5}$). Unlike projections from FP and vmPFC, this trend was less evenly distributed along a gradient from sites 1–6, as the projections are concentrated in site 6. The final projection pattern was a single-peak nonmonotonic gradient centered at site 4. The average input strength from dlPFC, vlPFC and dmPFC increased from site 1 (~1%) to site 4 (>5%) and then decreased from site 4 to site 6 (~2%). The change across sites was more gradual than that in the two monotonic trends. The between-site difference was statistically significant (Kruskal-Wallis $H = 20.18$, p<0.01). Interestingly, OFC areas did not show a trend across the rACC (*Figure 6D*). Rather, the OFC projection was strongest to site 5 (8%), 1 (5%), and 4 (4%) and less so to sites 2 (2%) and 6 (1%).

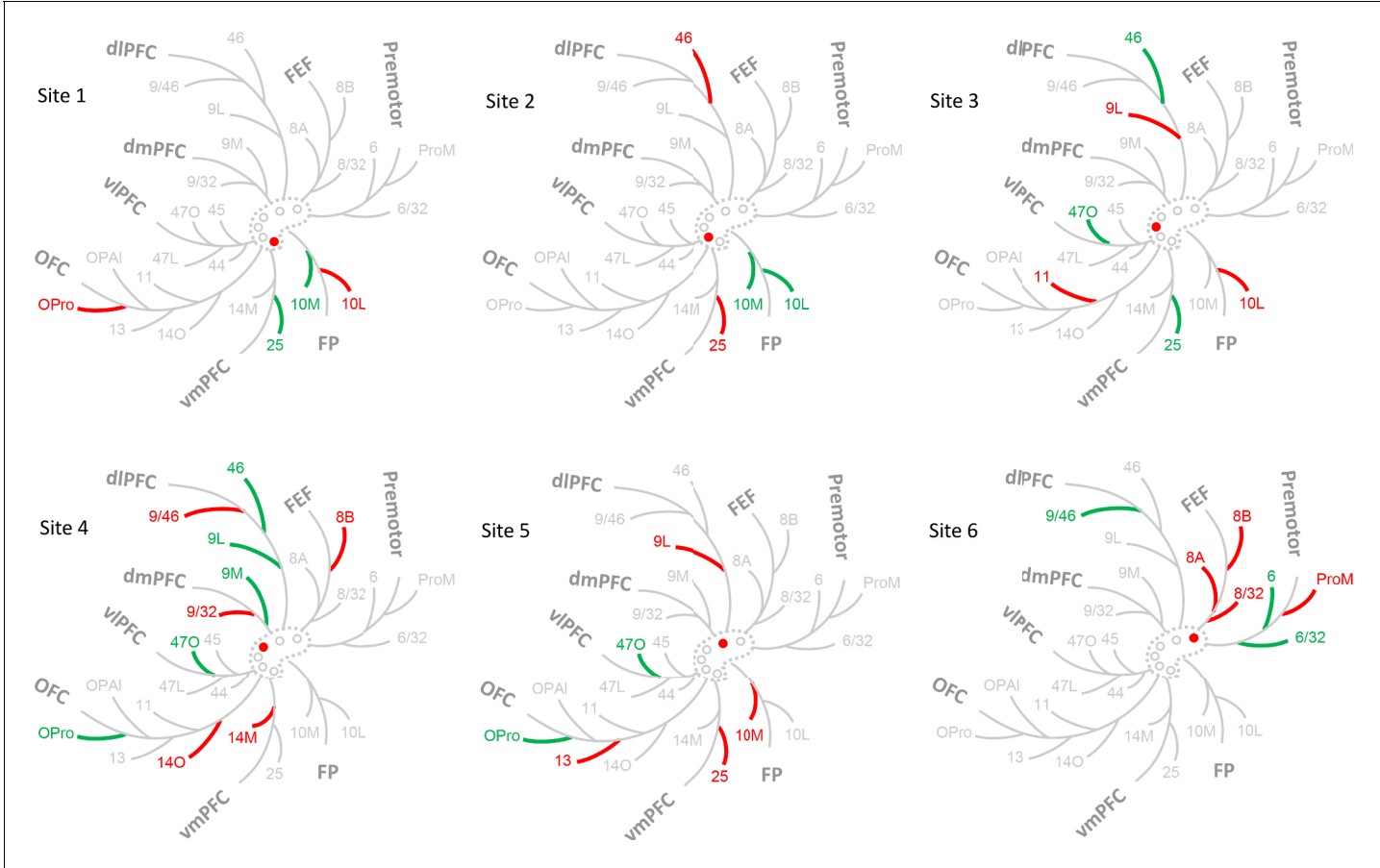

**Figure 5.** Schematic illustration of the FC regions with strong projections in each case. The dashed contour represents the rACC and the circles injections of cases 1–6. The filled circle marks the case being shown. Colored branches represent brain areas that accounted for 75% cell counts in each case (green = adds up to 50%; green + red = adds up to 75%). The most extensive FC regions with strong input was found in case 4.

DOI: https://doi.org/10.7554/eLife.43761.008

## Summary of projection patterns for sites 1–6

FC inputs to six sites in the rACC varied with respect to strength and regions of origin (*Table 2*). Sites 1 and 2 received the strongest (50%) inputs from two FC areas (regions of origin: vmPFC and FP); site 3 received the strongest inputs from three areas (vmPFC, vlPFC and dlPFC); site 4 received the strongest inputs from five areas (OFC, vlPFC, dmPFC and dlPFC); site 5 received the strongest inputs from two areas (OFC and vlPFC); and site 6 received the strongest inputs from three areas (dlPFC and premotor cortex). Across sites, the projection strength of different regions formed three spatial patterns: vmPFC and FP showed decreasing projection strength from site 1 to 6; FEF and premotor cortex showed increasing projection strength from site 1 to 6. The third pattern was a non-monotonic gradient formed by projections from dlPFC, dmPFC and vlPFC, with a single peak centered at site 4. To test whether the projection patterns found above were robust against the variability of 1. injection size, 2. sites above and below the cingulate sulcus, and 3. data reported here and from the literature, we analyzed three additional cases.

## Projection patterns of the control cases

All three control sites were located close to site 5. Site 5a served as a control for the variability of injection size; site 5b, the variability of sites dorsal or ventral to the cingulate sulcus; and site 5c, the variability across studies. Site 5a was a large injection and, consistent with its size, had a higher number of labeled cells than all the other sites (*Figure 7—figure supplement 1B*). However, despite the greater number of overall labeled cells, the degree-centrality of site 5a was similar to that of site 5.

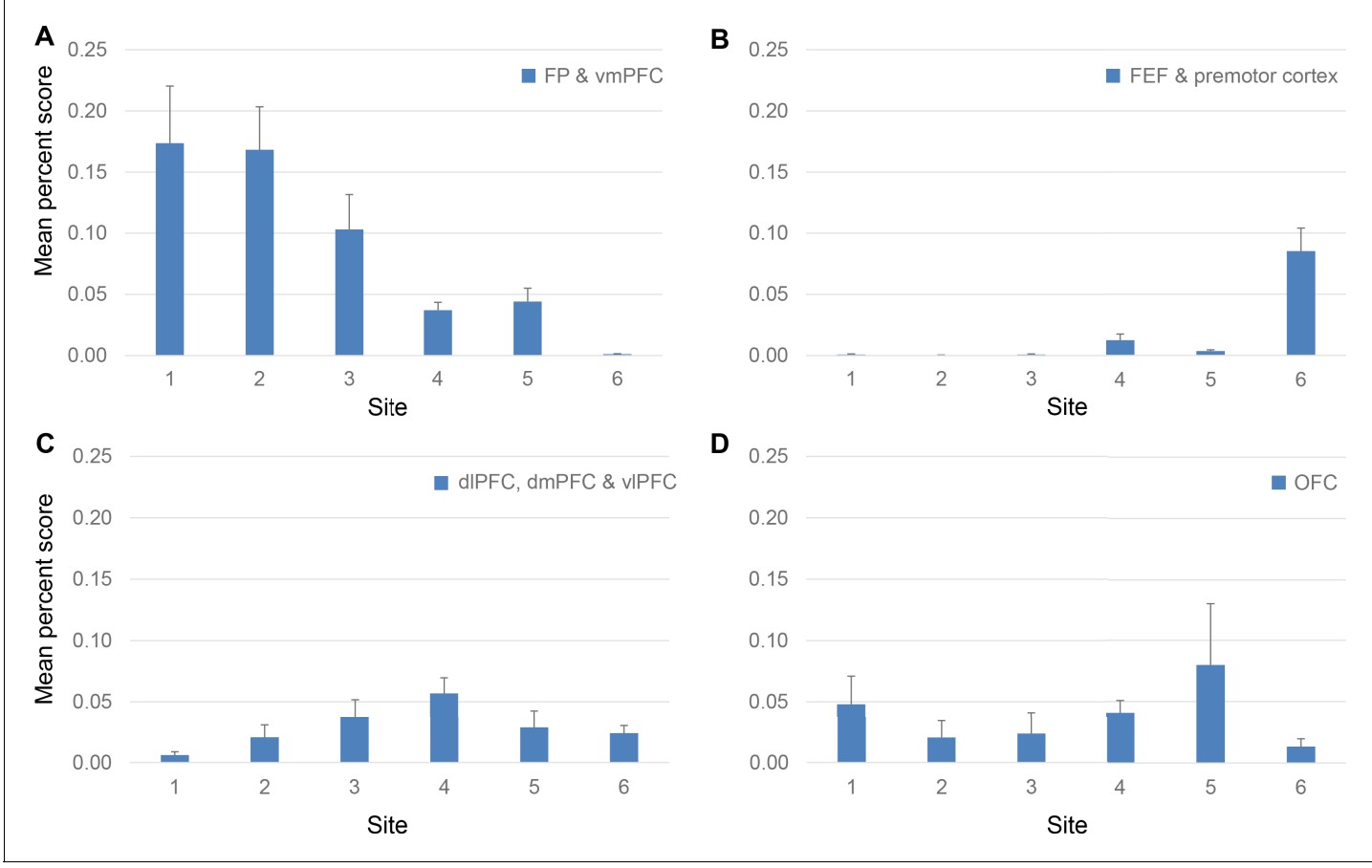

**Figure 6.** Projection strength from different FC regions at each site. Percent scores of inputs from (**A**) FP and vmPFC, (**B**) FEF and premotor cortex, (**C**) dlPFC, dmPFC and vlPFC and (**D**) OFC are shown in separate panels. In each panel, the percent scores of areas in the corresponding regions were averaged. The mean and standard error across areas are shown for each site. The mean percent score of FP and vmPFC was greater at sites 1–3 than at sites 4–6; that of FEF and premotor cortex was lower at sites 1–3 than at sites 4–6. The mean percent score of dlPFC, dmPFC and vlPFC gradually increases from site 1–4 and decreases from site 4–6. There was no consistent pattern in the OFC percent scores across sites.
DOI: https://doi.org/10.7554/eLife.43761.009

Site 5a received 50% of its inputs from two areas, 13 and 47O. Three areas, OPAl, 47L and 46 contributed the next 25%, for a total of five areas comprising 75% inputs (*Figure 7A*, left panel). In comparison, the number of areas contributing 50% and 75% inputs to site 5 were two and six (*Figure 4*). The distribution of inputs among FC regions also showed similarity between sites 5a and 5. The FC regions contributing 50% inputs to site 5a were OFC and vlPFC, the same regions that contributed 50% inputs to site 5. The next 25% of inputs to site 5a were from OFC, vlPFC and dlPFC (*Figure 7B*, left panel), compared to OFC, vmPFC, dlPFC and FP for site 5.

Site 5b was a control for the variability of inputs to the dorsal bank of the cingulate sulcus, a location frequently recorded in NHP electrophysiology experiments and demonstrating diverse functional associations. We examined whether inputs to site 5b differed from those to site 5. Site 5b received 50% inputs from four areas, 47O, 47L, 6L and 8B. Four areas contributed the next 25%: 44, 45B, 46 and 8A, for a total of eight areas comprising 75% inputs (*Figure 7A*, right panel). The FC regions contributing 50% inputs were vlPFC, FEF and premotor cortex, and the next 25% were vlPFC, FEF and dlPFC (*Figure 7B*, right panel). Compared with site 5, more areas contributed to 50% and 75% of the total inputs to site 5b. However, these areas were concentrated in the vlPFC and motor control regions, fewer regions compared to site 5.

The analysis of control case 5c demonstrated that regardless of the tracer type or experimental procedure used by different research groups, injections at the same location resulted in consistent labeling patterns. Site 5c matched most closely with a case previously reported in the literature

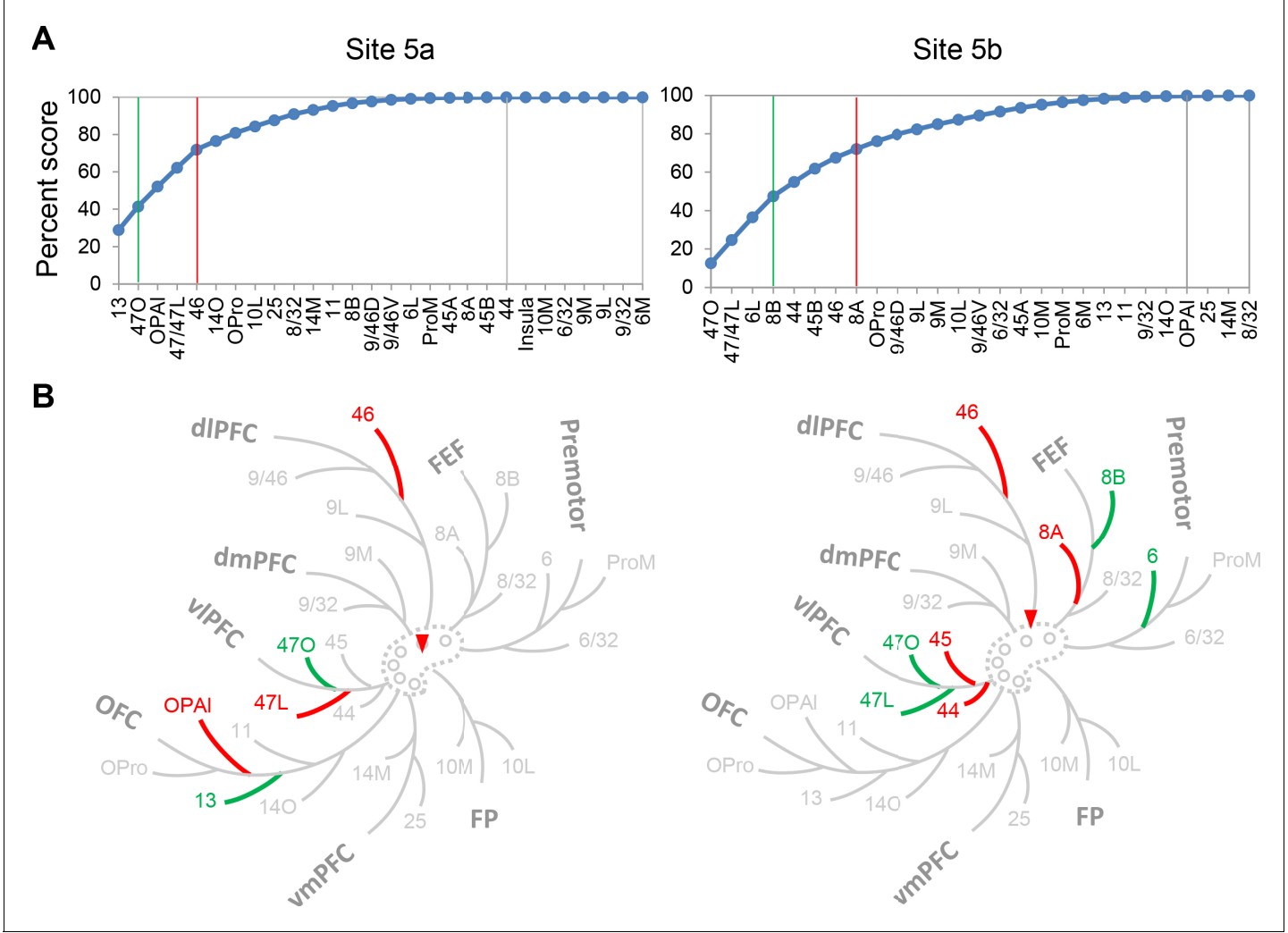

**Figure 7.** Projection patterns of the control cases 5a and 5b. (A) Cumulative percent cell count across areas. Cutoff remarks: green line = 50%, red line = 75%, gray line = 100%. Areas after 100% are in a random order. (B) Illustration of the FC regions with strong projections following the same schema as in *Figure 5*. Red triangle marks the location of the injection.
DOI: https://doi.org/10.7554/eLife.43761.010

The following figure supplements are available for figure 7:

**Figure supplement 1.** Percent scores of retrogradely labeled cells were not dominated by the size of the injection site or that of the FC areas.
DOI: https://doi.org/10.7554/eLife.43761.011

**Figure supplement 2.** Projection patterns from the FC areas to control site 5 c.
DOI: https://doi.org/10.7554/eLife.43761.012

(*Morecraft et al., 2012*). Compared to the injection of case 3 in *Morecraft et al. (2012)*, site 5c was within 1 mm more rostral (see our *Figure 7—figure supplement 2A* vs. their *Figure 7C*, the genu of the corpus callosum as landmark). The cell labeling patterns of these two cases were similar in the matching coronal sections (*Figure 7—figure supplement 2A* vs. Figure 7C in *Morecraft et al., 2012*). Both cases showed dense labeling in areas 10, 9M, 32, 24, 8B and caudal OFC. We further quantified the labeled cells of site 5c for comparison with the cases in this study. The sites 5c and five showed similar degree-centrality. Site 5c received 50% inputs from three areas (two for site 5): 9M, 25 and 47O. Five areas (four for site 5), 47L, 46, 10L, OPro and 10M contributed the next 25%, for a total of eight areas comprising 75% inputs (*Figure 7—figure supplement 2B*). The FC regions contributing 50% inputs to site 5c were vmPFC, vlPFC and dmPFC, and the next 25% were FP, OFC,

vlPFC and dlPFC (*Figure 7—figure supplement 2C*). All of these regions except dmPFC contributed to 75% of inputs to site 5.

## Comparison of the inputs to site 4 with the composition of inputs to areas 32 and 24

Site 4 stands out as having a uniquely diverse input pattern as compared to the other sites. It showed the highest number of areas that contributed 50% and 75% of its inputs; these areas were distributed among the highest number of FC regions. Moreover, site 4 was in the intermediate zone of two projection gradients and was the peak center of a third gradient. These connectivity patterns suggested that site 4 is a hub in the FC-rACC network. However, because site 4 was at the junction of areas 32 and 24, an important question was whether its input pattern could be explained by a composition of inputs to areas 32 and 24. If the projection pattern changed gradually from areas 32 to 24, then the strength of the major FC projections to site 4 would fall between the strength of these FC projections to areas 32 and 24. Moreover, if the projection pattern of site 4 was a composition of inputs to areas 32 and 24 but one area contributed more than the other, then this inequality should be present consistently across FC areas. In other words, one would expect to see a closer and consistent approximation of projection strength between site 4 and either area 32 or 24. We tested these possibilities with additional analyses.

We compared the strength of projections to site 4 with the average strength of projections to area 32 (averaged percent scores of sites 1–3) and those to area 24 (averaged percent scores of sites 5 and 6) (*Figure 8A*). Areas 14O, 11, 47L, 9/46D, 9L, 9M, 9/32 and 8A showed stronger projections to site 4 than to either area 32 or 24 (*Figure 8A*). Moreover, 14 out of the 27 FC areas showed more similar projection strength between site 4 and area 24 than site 4 and area 32 (*Figure 8A*). We further compared the projection pattern of site 4 with different weighted averages of the connection patterns of areas 32 and 24. In the test for linear combinations, the similarity between normalized connectivity patterns was measured by the symmetric Kullback-Leibler distance. The best fit to site 4 was a combination of approximately 2/3 of area 24 and 1/3 area 32. However, the distance measures were indistinguishable from a situation where we replaced one of the two areas' connectivity patterns with a uniform distribution over the FC areas (*Figure 8B*). In addition, the optimum mixture was inconsistent across the different FC areas. In many cases the connectivity of site 4 was higher/lower than predicted by the weighted average model (*Figure 8C*). These results demonstrated that the input pattern of site 4 was not a simple linear mixture of the patterns of areas 32 and 24. To test non-linear mixtures of areas 32 and 24, we generated 1000 mixed patterns using randomly sampled, varying weights across FC areas. We compared the degree-centrality and entropy between the mixed pattern and the pattern of site 4. Among the 1000 random samples, only 38 showed five or more FC areas contributing 50% inputs ("degree-centrality at 50%"), and 63 showed 10 or more FC areas contributing 75% inputs ("degree-centrality at 75%"). Moreover, only 117 of the 1000 samples showed greater entropy than that of site 4. In contrast, when we replaced one of the two areas' connectivity patterns with a uniform distribution, more than 600 samples showed degree-centrality $\geq 5$ at 50%, and more than 900 showed degree-centrality $\geq 10$ at 75%. More than 900 samples showed greater entropy than that of site 4. Thus, the hub properties of site 4 was better predicted by the mixture with a uniform distribution than the combination of patterns of areas 32 and 24. Taken together, these results suggested that the hub properties of site 4 did not occur due to the mixing of connectivity patterns of areas 32 and 24.

Finally, in a reconstructed 3D model, we verified that the cortical locations of retrogradely labeled cells of site 4 was not a composition of the spatial patterns of labeled cells of its neighboring sites 3 and 5 (*Figure 8D*). In the reconstructed 3D model, site 4 showed a high number of cells covering the caudal dlPFC and the principle sulcus, consistent with the high projection strength found with areas 9L, 8B and 9/46D in *Figure 8A*. Neither of its neighboring sites 3 and 5 showed high coverage of cells in these parts of cortex.

## Convergent probabilistic tracts from the FC to the rACC in dMRI

The FC of NHP dMRI images was parcellated into 27 areas that corresponded to the 27 areas used in the tracing analysis (*Figure 9A*). Each area was used as a seed mask. Areas 24 and 32 were combined as the target mask. Probabilistic streamlines from the different seeds terminated in partially

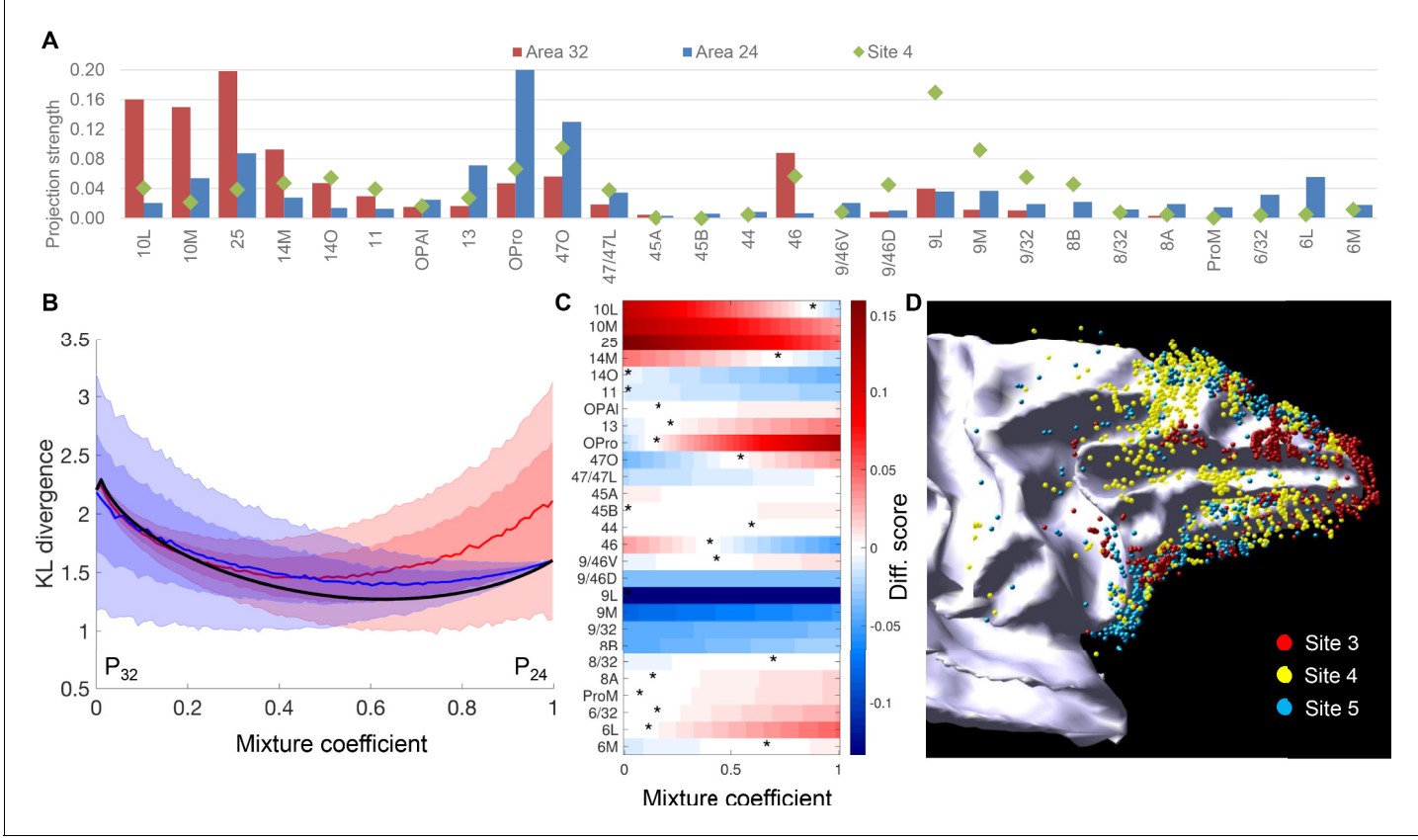

**Figure 8.** Comparison of projections to site 4 with those to areas 32 and 24. (**A**) Projection strength across FC areas. Site 4: projection strength = percent cell count; area 32: projection strength = averaged percent cell count across sites 1–3; area 24: projection strength = averaged percent cell count across sites 5 and 6. (**B**) Comparison of site 4 cell counts to mixtures of connectivity profiles of nearby areas 32 and 24. The comparison uses the symmetric Kullback-Leibler divergence as a measure of distance between the normalized cell counts histograms. Although the distance is minimized for a mixture of approximately 2/3-1/3, this mixture is indistinguishable from mixing the distributions of areas 32 or 24 with a uniform distribution (shaded areas show mean +1 std (dark) and 2std (light) calculated with 10000 histograms of a uniform distribution). (**C**) Colors show the difference between the percent counts of site 4 compared to a linear mixture of areas 32 and 24. Optimal mixtures for individual target areas, corresponding to the color white, shows that the optimal mixture coefficient is variable across areas, but seven areas do not have an optimum (all red or all blue). (**D**) A 3D model of retrogradely labeled cells of sites 3, 4 and five on the white matter surface.

DOI: https://doi.org/10.7554/eLife.43761.013

overlapping regions in the target mask. Each streamline was a probabilistic estimation of the path that connected a seed voxel and a voxel in the ACC. As an example, *Figure 9B* shows two seed masks from areas 11 and 46, and *Figure 9C* illustrates the voxels where streamlines from the two seed masks terminated in the ACC in one monkey. A subgroup of the streamlines from both areas targeted the same voxels (shown in orange in *Figure 9C*). We identified the location of highest convergence, that is the voxel receiving streamlines from the greatest number of seeds. A convergent-connectivity value was calculated for each voxel in the target mask, approximating the number of areas with high density of streamlines to that voxel. Consistently across seven monkeys, the highest convergent-connectivity value was located at the rostral edge of the cingulate sulcus (*Figure 9D*). This was in a similar location as site 4 in the tracing experiments (see *Figure 1A*). There was some individual variability on the dorsal-ventral axis. In three animals, the highest convergent-connectivity value was just dorsal to the cingulate sulcus and in four animals it was just ventral to the sulcus.

We applied the same tractography method to human dMRI data. The human FC was parcellated into 25 areas following *Petrides and Pandya (1994)*. This anatomical division was developed to maximize architectonic correspondence between human and NHP frontal areas (OPro and OPAl were not clearly defined in the human parcellation by *Petrides and Pandya, 1994*, and were thus not included in our human dMRI analysis). Similar to the NHP analysis, areas 32 and 24 were

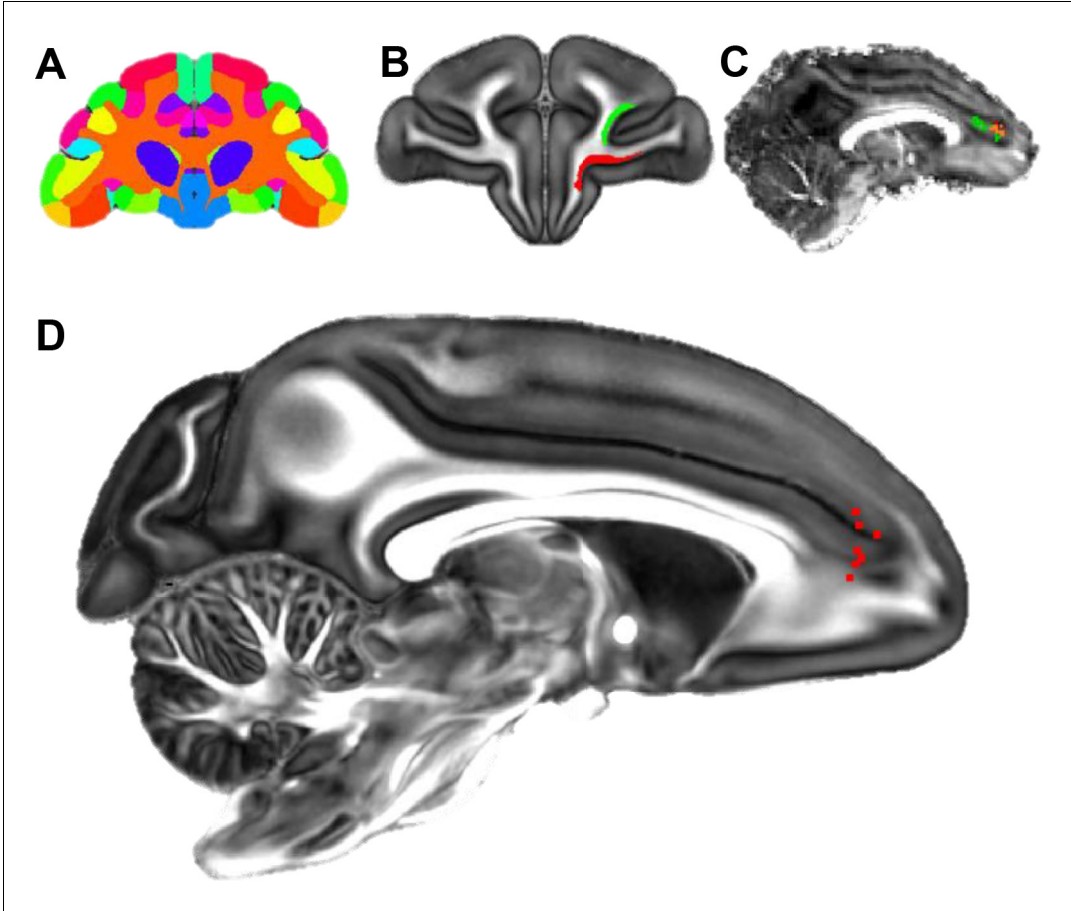

**Figure 9.** Localization of the hub region in the monkey rACC using dMRI tractography. (**A**) A coronal section illustrating the Paxinos atlas in the dMRI space provided by Duke University. (**B**) A coronal section of the atlas brain showing two example seed masks for areas 11 (red) and 46D (green). (**C**) A sagittal section of an individual monkey brain showing the probabilistic streamline terminals in the rACC, separately for the seeds in area 11 (red) and 46D (green). Voxels with overlapping terminals were in orange. (**D**) A sagittal section showing the localized hub in seven individual monkeys. Each red dot marks the center of the hub region in one monkey. The center of the hub was defined by the voxel with the highest weighted-sum of probabilistic streamlines from all 29 seeded areas.
DOI: https://doi.org/10.7554/eLife.43761.014

The following figure supplement is available for figure 9:

**Figure supplement 1.** Difference in the tract strength pattern due to seeding procedures.
DOI: https://doi.org/10.7554/eLife.43761.015

combined as the ACC target mask. Probabilistic streamlines were generated from each seed to the target. A convergent-connectivity value was calculated for each voxel in the target mask. The results demonstrated that, same as in the NHP results, streamlines in each subject converged in the rACC. The highest convergent-connectivity value was consistently located in the rostral part of the rACC for all subjects (*Figure 10*). The geometric center of the individual results was at the genu of the cingulate gyrus. This region was spatially approximate to site 4 in the NHP tracing study. Despite the morphological difference of cingulate sulcus between human and NHP, the geometric center of the results in *Figure 10* was at the rostral edge of the human cingulate sulcus, based on the Mai human atlas (*Mai et al., 2008*). Interestingly, as in the NHP data, there was little individual variance in the rostrocaudal location of the convergence. However, similar to the NHP results, there was some individual variation in the dorsal-ventral axis. Importantly, as seen in NHP data, about half of the subjects had the voxel of highest convergence above the genu, and the other half below the genu.

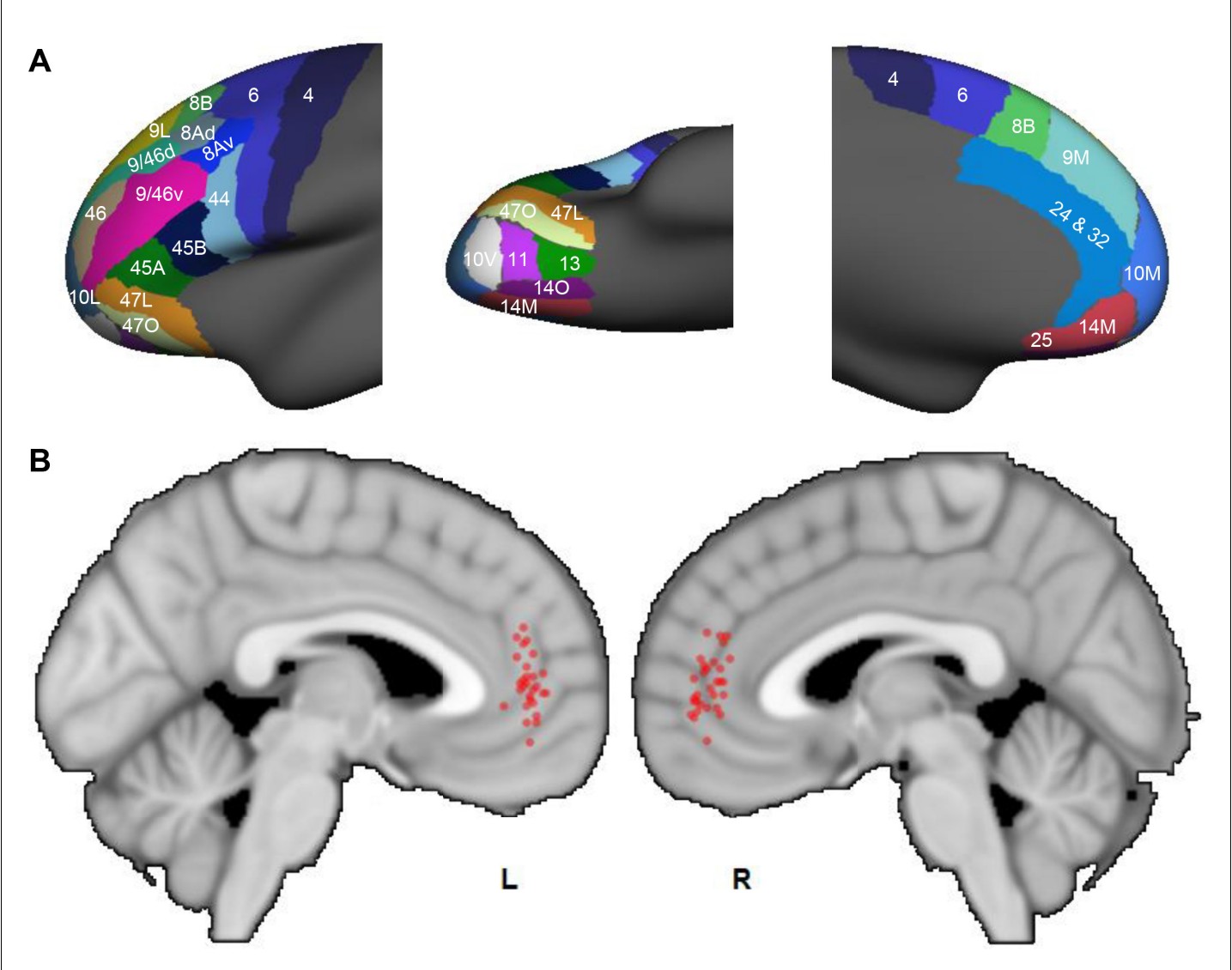

**Figure 10.** Probabilistic streamlines converging in the rACC in human dMRI. (**A**) Parcellation of the FC areas on the *fsaverage* template (FreeSurfer 4.5), following *Petrides et al. (2012)*. The FreeSurfer labels are available in *Supplementary file 1*. (**B**) Sagittal sections showing the localized hub across individuals. Each red dot marks the center of the hub region in one subject. The center of the hub was defined by the voxel with the highest weighted-sum of probabilistic streamlines from all seeded FC areas.

DOI: https://doi.org/10.7554/eLife.43761.016

## Discussion

In this study we mapped the FC inputs to different subregions within the rACC and determined the strength of connections from each FC region and areas within each region. Site 4 showed the highest number of regions and areas within regions that contributed the strongest inputs, thus showing the highest degree-centrality. Site 4 was in the intermediate transition zone of two projection gradients across the six sites. It was also at the peak center of a third, nonmonotonic, gradient. Together with the degree-centrality, these connectivity patterns suggested that site 4 is a hub within the rACC. The dMRI tractography in the NHP demonstrated that streamlines from the FC converged in a location comparable to site 4. Using the same tractography methods in human, we found that streamlines from FC also converge in a similar position in the human rACC. Thus, using a cross-species, multimodal approach we located a hub in the rACC in NHPs and, in a similar position, a likely hub in human rACC.

Qualitatively, our results were overall consistent to previous studies in the literature that demonstrated vmPFC, dlPFC and caudal OFC projections to the rACC (*Vogt and Pandya, 1987*; *Barbas and Pandya, 1989*; *Carmichael and Price, 1995*; *Barbas and Pandya, 1989*; *Morecraft et al., 2012*). However, in this study, we quantified the results and compared the relative projection strength for the different inputs. We compared descriptive and quantitative analysis in one case (5 c), which had an injection site similar to case 3 in *Morecraft et al. (2012)*. Although the published data appeared to have a greater number of labeled cells, both cases showed similar labeling patterns. When we quantified the results and tested for the degree-centrality, the results showed that 50% of the inputs to site 5 c were derived from only three areas. This analysis demonstrated the importance of quantification for determining the relative strength for connections. The degree-centrality was found invariant to injection size (control case 5a) or whether the injection was in the dorsal or ventral bank of the cingulate sulcus (control case 5b).

## Species homologies

Comparative studies have utilized spatial location, cytoarchitectonics, connectivity pattern and functional activation to relate homologous areas between human and NHP. The numerical designations of FC areas in early human and NHP atlases (*Brodmann, 1909*; *Walker, 1940*) have been found most consistent in the dlPFC, while major disagreement exists in ventral and orbital areas (*Barbas and Pandya, 1989*; *Petrides et al., 2012*). Such inconsistency has been further studied and alleviated based on fine-grained cytoarchitectonics and connection patterns (*Carmichael and Price, 1995*; *Mars et al., 2018*; *Ongür et al., 2003*; *Petrides and Pandya, 1994*; *Petrides et al., 2012*). The atlases used in this study (*Paxinos et al., 2000*; *Petrides and Pandya, 1994*) have incorporated this literature, with a particular focus on reconciling terminology inconsistencies between NHP and human (*Petrides et al., 2012*). There are three major divisions of the ACC in NHPs (*Morecraft and Tanji, 2009*): the sACC (area 25), the pregenual ACC (pACC, area 32 and rostral part of area 24) and the midcingulate (caudal part of area 24). The pACC together with the rostral part of the midcingulate is also referred to as the dACC. In human neuroimaging studies, the definition is less precise. Different from the NHP terminology, the human dACC does not include parts of the pACC but only the rostral part of the midcingulate. The pACC is often referred to as the rACC. To avoid confusion, in this study we use rACC to refer to pACC in both NHPs and humans.

An important issue is whether the hub is in a homologous region within the rACC. We base our assessment on the surface features and cytoarchitecture of the ACC. The cytoarchitectonic regions of the human cingulate cortex have been found broadly consistent with the NHP region (*Vogt, 2009a*; *Vogt et al., 2006*; *Vogt et al., 1995*). The two areas in this study, 32 and 24, occupy the genu of the cingulate cortex in both species. The hub region found by tract tracing was at the conjunction of NHP areas 32 and 24. In human, this conjunction region is located at the rostral-most edge of the cingulate sulcus (*Vogt et al., 1995*). Thus, using this surface feature as guidance, we compared the human dMRI result (*Figure 10*) with a human brain atlas (*Mai et al., 2008*) to verify that the highest convergent connectivity was near the conjunction of areas 32 and 24.

## Translation of results between species via dMRI

Comparing two connectivity modalities (tracing and dMRI) in different species (NHP and humans) is problematic. We therefore first compared the tracing results with dMRI scans in NHPs, two of which were in the same animals that received the tracer injection. This comparison was central for determining how accurately the dMRI method was able to replicate the anatomic findings. The most direct way to compare the methods was to place the dMRI seeds at the location of each injection site and follow the streamlines to the FC. The results showed that streamlines from these seeds could not be followed to the cortical areas, but rather remained within the cingulum bundle (*Figure 9—figure supplement 1*). This problem was due to the fact that fibers are highly aligned in the cingulum bundle along the anterior-posterior axis. Consequently, the diffusion signals were strongly anisotropic towards the anterior-posterior direction, while being disproportionately weak in the other directions. Thus, streamlines from the ACC seeds had very low probability of leaving the cingulum bundle (*Figure 9—figure supplement 1A & B*). The low number of streamlines reaching different FC areas resulted in low statistical power for measuring the connectivity pattern. To address this problem, we placed seeds in the FC areas which resulted in sufficient number of streamlines

that correctly reached the ACC target. The connection strength calculated from these streamlines showed a better match with the projection strength from the tracing data than the results with seeds in the ACC (*Figure 9—figure supplement 1C*).

## The characterization of a network hub by anatomical projection patterns

While the concept of hubs emerged in early discussions of brain networks (*Mesulam, 1990*), recent advances of network analysis have formalized the definition of hubs using graph theory (*Sporns, 2011*). Theoretically, a hub is the node of the highest degree in a network, that is. with unusually high connections with the other nodes. In modularized networks as those in the brain, the hub facilitates communication between functional modules. Neuroimaging studies have identified the rACC as a hub of the brain's global network (*Buckner et al., 2009*; *Hagmann et al., 2008*). However, the rACC is a large region under the scope of anatomical analysis. Anatomical inputs vary across subregions within the rACC. The hub-like connectivity observed in neuroimaging studies may simply reflect the sum of connections over all of rACC's subregions. The question is then whether a hub exists as a confined subregion within the rACC.

In this study, site 4 showed two defining features of a hub: high degree-centrality and a position in the network that could facilitate cross-module integration. High degree-centrality was reflected by the number of areas with strong projections to this site (*Figures 3–5*). To demonstrate cross-module integration, we defined functional modules in an empirical manner. The standard graph theory definition requires all-to-all connectivity measured between FC areas (*Sporns, 2011*). However, this approach is impractical with tract tracing experiments. We used functional regions of the FC to approximate functional modules. Site 4 marked the most integrative zone in the rACC where inputs from the largest number of functional regions converge: 1. Site 4 was in the central location of two projection gradients. One is formed by inputs from vmPFC and FP (*Figure 6A*). These were regions associated with emotion and decision making (*Joyce and Barbas, 2018*; *Piray et al., 2016*; *Tsujimoto et al., 2010*). They had the strongest inputs to sites 1–3, less so to sites 4 and 5, and the least to site 6 (*Figure 6A*). Inputs from the FEF and the premotor cortex were part of the second gradient. These were motor control regions that had the strongest projections to site 6, less so to sites 4 and 5, and the least to sites 1–3 (*Figure 6B*). At each end of the two gradients, the inputs were predominantly associated with emotion- and motor-related functions, respectively. Site 4 was in the intermediate transitioning zone of both gradients. Input strengths were more similar between gradients at this site. 2. A third gradient peaked at site 4. Input strength from regions associated with cognition gradually increased from site 1–4 and decreased from site 4–6 (*Figure 6C*). This peak of inputs at site 4 positioned this site to maximally interface inputs between cognition, emotion and motor control. Therefore, site 4 enables communication between all three functional modalities. Together with the high degree-centrality, the integrative nature of site 4four suggests that it is a hub in the frontal cortex.

## A hub or inputs reflecting the junction of two cytoarchitectural areas?

An alternative to the hub hypothesis is that site 4 was at the border of areas 32 and 24, and thus, its diverse inputs could be a result of a composition of inputs to areas 32 and 24. The analyses of projection strength across FC areas and the spatial distribution of labeled cells argue against this alternative (*Figure 8*). 1. If the inputs to site 4 were a composition of the inputs to areas 32 and 24, there would be none or few of the frontal areas projecting to site 4 with higher or lower strength than their projections to both areas 32 and 24. In contrast, we found 8 FC areas that showed stronger projections to site 4 than to either area 32 or 24 (*Figure 8A*). 2. The 'composition' hypothesis indicates that the strength of projections to site 4 would be consistently more similar to that of one cytoarchitectonic area than the other, depending on which area site 4 covers more extensively. However, we found this similarity random, that is about a half of the FC areas projecting with similar strength to site 4 and area 32 than site 4 and area 24 (*Figure 8A*). 3. The 'composition' hypothesis also indicates that the strength of projections to site 4 would be predicted by a weighted average of the connectivity patterns of areas 32 and 24. In the linear mixing analysis, although a model with 1/3 of the pattern of area 32 and 2/3 or the pattern of 24 was a better fit to site 4 than either area on its own, such fit was indistinguishable from mixing either area with a uniform distribution of connections to the FC areas (*Figure 8B*). In addition, the optimal mixture was variable across the FC areas

(*Figure 8C*). In the non-linear mixing analysis, the combined input patters of areas 32 and 24 poorly predicted the degree-centrality and entropy of site 4. Such predictions were better when using a mixture of either area and a uniform distribution. 4. In the reconstructed 3D model of the retrogradely labeled cells, site 4 showed a high number of cells in the caudal dlPFC and the principal sulcus, which were only sparsely covered by cells projecting to the neighboring sites 3 and 5 (*Figure 8D*). Thus, the connectivity profile of site 4 cannot be explained by a composition of connectivity profiles of its neighboring sites. Based on the collective results, we propose that inputs to site 4 are best understood via the high degree-centrality and cross-domain functional integration at site 4. This cannot be predicted by the location of this site among the cytoarchitectonic areas.

## Methodological limitations

A main limitation of the current study is that experiments were performed in different animals. It is therefore difficult to assess individual variability of the input patterns for each site. Ideally, each animal would receive tracer injections along the rACC and this would be repeated across several animals. Unfortunately, technically this would be difficult to accomplish without, for example, tracer contamination. A second limitation is that the analyses cannot completely exclude the possibility that within-region microcircuitry heterogeneity leads to the observed patterns of a hub. The microstructure of a region between two cytoarchitectonic areas might involve a complex spatial composition of neurons from each area. The inputs to these neurons form heterogeneous microcirctuitries that are not visible in the tracing experiments. Thus, when observed as a whole, the region containing these neurons may appear as a hub.

## Implications for ACC functions

The rACC hub provides an alternative view to the dichotomous interpretation of ACC functions, such that emotional and cognitive influences affect the ventral and dorsal ACC separately (*Bush et al., 2000*). Classically, the ventral ACC is attributed with 'limbic' functions, for example visceral responses, emotion, and memory (*Buckner et al., 2008*; *Etkin et al., 2015*; *Mayberg et al., 1999*; *Papez, 1995*; *Vogt et al., 1992*). The dorsal ACC is linked with executive control, typically for choosing from conflicting actions and monitoring behavioral outcomes (*Botvinick, 2007*; *Holroyd and Coles, 2002*; *MacDonald et al., 2000*; *Pardo et al., 1990*). However, this coarse dichotomy cannot entirely account for the complex activation patterns of the ACC across an increasing body of experiments. Recent theories have acknowledged the influence of reward and economic evaluation on the executive component of the dorsal ACC function (*Botvinick and Braver, 2015*; *Kolling et al., 2016*; *Shenhav et al., 2016*), even though reward processing is classically thought of as a ventral ACC/vmPFC function. The effect of cognitive regulation over emotion in the rostral/ventral ACC has also been addressed in the literature on fear extinction (*Etkin et al., 2015*; *Klumpp et al., 2017*).

The rACC hub suggests an alternative view on the functions of ACC subregions. In contrast to associating each subregion with one function to support serial computation, the ACC may be better understood through the functional integration by its subregions. Neuroimaging studies have reported high degree of structural and functional connectivity between the rACC and the rest of the brain (*Buckner et al., 2009*; *Hagmann et al., 2008*). While such studies often treated the rACC as a single large region, our results demonstrate that the integration can be carried out in a precise subregion. Site 4 is not simply a conjunction between the ventral and dorsal ACC, but a hub that routes information across functional modules. The inputs to the hub contain strong projections from vlPFC, dlPFC and dmPFC. These regions are critical for higher cognition, such as social behavior (*Stalnaker et al., 2015*), decision making (*Kable and Levy, 2015*; *Sakagami and Pan, 2007*; *Wallis, 2007*), learning (*Atlas et al., 2016*; *Schuck et al., 2016*), attention (*Corbetta and Shulman, 2002*; *Uddin, 2015*) and working memory (*D'Esposito and Postle, 2015*; *Miller and Cohen, 2001*). Thus, the hub can route the outputs of higher cognitive functions to emotion and executive processing within the ventral and dorsal ACC. The hub may be uniquely positioned for evaluating and arbitrating between these processes (*van den Heuvel and Sporns, 2013*). The detailed mechanisms may be further investigated through connections of the hub and the other subregions of the ACC.

## Potential hubs in other subregions of the rACC

The results in this study did not suggest that only one hub exists in the rACC. Despite the advantage of measuring projection strength at cellular level, this study – and generally tract tracing studies – have a limited number of cases. Unlike neuroimaging studies that can identify all hubs in the global network of the brain, we can only test for hub locations among the available injection sites. Thus, the experiment did not exhaustively cover the all ACC regions. It is possible, therefore, that additional hubs are located elsewhere. The dorsal bank of the cingulate sulcus at the rostro-caudal level between sites 5 and six is a region of particular interest as it is often the site for NHP electrophysiology experiments. These studies demonstrate the functional diversity of this region. To determine whether this area had the characteristics of a hub, we analyzed case 5b (*Figure 7*). We found that 50% of the inputs to site 5b were contributed by 4 areas and 3 FC regions (*Figure 7A & B*, right panel). The degree-centrality measured by FC regions was similar between site 5b and sites 1, 2 and 6. Although site 5b is unlikely to be a hub, it showed connectional heterogeneity with strong inputs from both vlPFC and premotor areas. This heterogeneity is consistent with findings in our other cases and in other cytoarchitectonic areas (*Luppino and Rizzolatti, 2000*), where inputs are derived from more than one functional region.

## Implications on the pathophysiology of psychiatric disorders

From a network perspective, most psychiatric disorders are seen as a consequence of network imbalance rather than localized deficits (*Menon, 2011*). This view conforms with and extends the disconnection hypothesis (*Catani and ffytche, 2005*): Damage to a hub region can cause disconnection between a wide range of functional modalities, and correspondingly, a spectrum of affective and cognitive disorders. The disconnection hypothesis is in line with the broadly observed rACC abnormality in various diseases, including major depression disorder (MDD) (*Mayberg et al., 1997*; *Pizzagalli, 2011*), obsessive-compulsive disorder (OCD) (*Beucke et al., 2014*; *Tadayonnejad et al., 2018*), attention deficit hyperactivity disorder (*Tomasi and Volkow, 2012*), and posttraumatic stress disorder (*Bryant et al., 2008*; *Kennis et al., 2015*; *Patel et al., 2012*). Based on the FC areas sending convergent inputs to the hub, dysconnectivity with the hub may be involved in the imbalance between goal directed control, emotion and higher cognition in these disorders. MDD and OCD both show treatment response in the rACC activity (*Chakrabarty et al., 2016*; *Fullana et al., 2014*; *Mayberg et al., 1997*; *O'Neill and Schultz, 2013*; *Pizzagalli, 2011*). The distinction in their pathophysiology lies in the type of networks involved: MDD engages the networks for self-reference and cognitive control (*Pizzagalli, 2011*), while OCD engages those for reward-driven and goal-directed behaviors (*Milad and Rauch, 2012*). The hub connects a majority of FC areas involved in the above networks, which makes it a site prone to damage in both disorders. The precise pattern of its anatomical connections provides important information for testing the disorder-specific disconnections.

## Acknowledgements

The anatomic tracing studies and dMRI analyses were supported by NIH/NIMH grants MH106435, MH045573 and U01-MH109589, and the UK Medical Research Council grant MR/L009013/1. Collection of dMRI animal data was supported in part by the Center for Functional Neuroimaging Technologies (P41-EB015896), Shared Instrumentation Grants S10RR016811, S10RR023401, S10RR019307, and the Human Connectome Project (U01-MH093765). We thank Anna Borkowska-Belanger for expert technical support. The authors declare no conflicts of interest.

## Additional information

### Funding

| Funder | Grant reference number | Author |
| --- | --- | --- |
| National Institute of Mental Health | MH106435 | Wei Tang<br>Ziyi Zhu<br>Julia F Lehman<br>Suzanne N Haber |

| National Institute of Mental Health | MH045573 | Wei Tang<br>Ziyi Zhu<br>Julia F Lehman<br>Suzanne N Haber |
| --- | --- | --- |
| Medical Research Council | MR/L009013/1 | Saad Jbabdi |
| National Institute of Mental Health | U01-MH109589 | Michiel Cottaar |
| NIH Blueprint for Neuroscience Research | U01-MH093765 | Giorgia Grisot<br>Anastasia Yendiki |

The funders had no role in study design, data collection and interpretation, or the decision to submit the work for publication.

## Author contributions
Wei Tang, Conceptualization, Data curation, Formal analysis, Investigation, Visualization, Methodology, Writing—original draft, Writing—review and editing; Saad Jbabdi, Formal analysis, Methodology, Writing—review and editing; Ziyi Zhu, Michiel Cottaar, Giorgia Grisot, Data curation, Methodology; Julia F Lehman, Data curation, Investigation, Visualization; Anastasia Yendiki, Data curation, Methodology, Writing—review and editing; Suzanne N Haber, Conceptualization, Resources, Data curation, Supervision, Funding acquisition, Investigation, Methodology, Writing—original draft, Project administration, Writing—review and editing

## Author ORCIDs
Wei Tang (iD) https://orcid.org/0000-0003-3550-4076
Giorgia Grisot (iD) http://orcid.org/0000-0003-4349-1201
Suzanne N Haber (iD) https://orcid.org/0000-0002-5237-1941

## Ethics
Human subjects: The human data were obtained from the publicly available Human Connectome Project database. All procedures conformed to ethical standards approved by the Institutional Review Board of Partners Healthcare. All human subjects have provided written informed consent.
Animal experimentation: All nonhuman primate experiments were performed in accordance with the Institute of Laboratory Animal Resources Guide for the Care and Use of Laboratory Animals and approved by the University Committee on Animal Resources at University of Rochester (Protocol Number UCAR-2008-122R).

## Decision letter and Author response
Decision letter https://doi.org/10.7554/eLife.43761.021
Author response https://doi.org/10.7554/eLife.43761.022

# Additional files
### Supplementary files
• Supplementary file 1. A cytoarchitecture-based surface parcellation of the frontal cortex.
DOI: https://doi.org/10.7554/eLife.43761.017
• Transparent reporting form
DOI: https://doi.org/10.7554/eLife.43761.018

### Data availability
All data analysed during this study are included in the manuscript and supporting files. FreeSurfer label files have been provided for Figure 10A. Data used from the MGH-USC Human Connectome Project (HCP) can be accessed following the creation of a free account (db.humanconnectome.org/data/projects/MGH_DIFF).

The following previously published dataset was used:

| Author(s) | Year | Dataset title | Database and Identifier |
|---|---|---|---|
| MGH-USC consortium of the Human Connectome Project | 2013 | MGH HCP Adult Diffusiondb. humanconnectome.org/data/projects/MGH_DIFF | HCP Database, MGH_DIFF |

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
