## [Decision Letter]

Thank you for submitting your article "A connectional hub in the rostral anterior cingulate cortex links areas of emotion and cognitive control" for consideration by *eLife*. Your article has been reviewed by three peer reviewers, and the evaluation has been overseen by David Badre, as Reviewing Editor, and Michael Frank as the Senior Editor. The reviewers have opted to remain anonymous.

The reviewers have discussed the reviews with one another and the Reviewing Editor has drafted this decision based on the reviewers' summary and comments to help you prepare a revised submission.

Summary:

The authors studied the inputs to six 'rostral anterior cingulate cortical' (rACC) areas with injection of tracers in two species of macaque monkeys and mapped the origin of projections from the frontal cortex. They then compared the data with findings obtained from diffusion MRI in monkeys and humans, using the injection sites as seeds to study pathways. They identified a non-monotonic connectional gradient and concluded that one site within rACC (their site 4) receives strong input from more areas than the rest, and so is a connectional hub.

Essential revisions:

The reviewers saw considerable value in this study. If verified, the gradient of connectivity and "hubness" of rACC would be an important and novel finding. Furthermore, there was an appreciation of the use of convergent methods between high resolution tracing and diffusion tractography for cross-species comparison.

However, the reviewers were also in agreement about the primary weakness of the study: The number of injections was highly limited and so this has left open several alternatives to a hub in rACC. For example, beyond some basic methodological concerns, there could be sites in the caudal regions left untested, such as dorsal ones, that would show a similar hub-like pattern, or there could be more than one region being conflated in what is being called rACC (site 4). Thus, the overall consensus was that more injections are needed in order to draw the broad conclusion that rACC is a connectional hub. Here are specific concerns along these lines from reviewers:

1) Three different tracers are used in the study, but it was not indicated which was injected where. Could the findings of strong labeling in more areas after a single injection in site 4 be attributed to a more efficient tracer, better uptake at the injection site, or a stronger antibody used in the processing of the tissue in each case? With one case per site and missing methodological detail, it is not possible to accept such a broad conclusion.

2) The conclusion that only one site in the cingulate cortex is a connectional hub is based on injection sites located mainly in the ventral bank of the cingulate sulcus (sites 5 and 6), whereas the site in rACC covers both the ventral and dorsal banks. Although it has been suggested that the fundus of the cingulate sulcus might be the dorsal limit of the cingulate cortex (Vogt et al., 2005), this dorsal limit has been challenged. There is functional and anatomical evidence that the dorsal bank of the cingulate sulcus belongs to the cingulate cortex (see Procyk et al., 2016 for review). For example, all electrophysiological studies demonstrating activity related to cognitive control and performance monitoring have recorded within the dorsal bank of the cingulate sulcus in monkey. So, can the authors exclude that another hub may exist in the dorsal bank of the cingulate sulcus (posteriorly to site 4, for example at the level of sites 5 and 6)?

3) Site 4 was at the intersection of areas 24 and 32 and sites 1-3 and 5-6 are located in areas 32 and 24, respectively (Results, first paragraph). Is it relevant that the injection in site 4 involves 2 areas and thus appears as a hub? The control case is intended to demonstrate that it is the location and not the size of the injection that matters, but the control case involves more posterior areas with less diverse profiles and number of connections. The fact that site 4 involves both areas 32 and 24 possess a problem since its "hubness" might be the result of the convolution of the connectivity profile of the two sites (3 and 5) with a high number of connections and diversity of connectivity profiles as Figure 4 indicates. The arguments against such an interpretation (subsection “Input patterns vs. cytoarchitectonic patterns”) are unconvincing. The first one does not provide evidence against the aforementioned interpretation. The second one assumes a simple sum of the connectivity profiles of two areas. But, this assumption is not necessary since there is connectional heterogeneity within areas (see for instance Luppino et al. work for the frontal cortex). In other words, there is not pure addition of connectivity profiles.

[Editors' note: further revisions were requested prior to acceptance, as described below.]

Thank you for resubmitting your work entitled "A connectional hub in the rostral anterior cingulate cortex links areas of emotion and cognitive control" for further consideration at *eLife*. Your revised article has been favorably evaluated by Michael Frank as the Senior Editor, David Badre as the Reviewing Editor, and three reviewers.

The manuscript has been improved but there are some remaining issues that need to be addressed before acceptance, as outlined below:

One of the main points raised previously was the claim that the "hubness" of site 4 could not reflect an injection involving 2 areas, that is, area 24 and 32. Though the control provided rules out an additive version of this account, there may be reason to doubt additivity of connectivity. Thus, reviewer 3 has suggested a set of additional control analyses to fully rule out this concern. These are appended in this reviewer's comments below. Reviewer 1 also has a suggestion regarding the strength of the conclusions that you should consider in your revision. In discussion, the reviewers were in agreement that both of these points should be addressed. Thus, I send this back to you for additional revision. The reviewers' specific comments follow.

Reviewer #1:

The authors addressed key points raised and concluded that due to technical limitations, there indeed could be other areas within the anterior cingulate region with strong connectivity resembling a 'hub'. In view of this conclusion, it would be consistent to also tone down the initial part of the discussion by removing words such as 'strongly' as in: "..the integrative nature of site 4 strongly supports that it is a hub in the frontal cortex..".

The manuscript varies in style of expression. For example, the first paragraph of Discussion reads like a section from Results. There is variation in the style of the references in the text, missing references from the reference list and misspellings of some names.

Reviewer #3:

I would like to thank the authors for their clarifications.

I would like to focus on one of the main points, namely, the claim of site 4 being a hub and that this is not explained by the fact that the injection involves 2 areas, that is, area 24 and 32.

The authors performed a nice analysis showing that site 4 connectivity cannot be explained by a linear addition of the pattern of connections of areas 24 and 32.

This is a basic demonstration of the fact that the assumption of "pure addition" of connectivity profiles does not necessarily hold. This was stated in my previous comment: "But, this assumption is not necessarily true since there is connectional heterogeneity within areas (see for instance Luppino et al. work for the frontal cortex). In other words, there is not pure addition of connectivity profiles."

So the authors indeed show that there is no pure linear addition as I indicated, but this is something that is distinct from claiming that the overlap might not contribute to the "overshoot" in the metrics that are used to classify sites as hubs.

A more direct rigid control analysis addressing the hubness of sites should focus on the criteria used for hubness and if they could be synthesized and to what extend they resemble site 4 hubness by mixing connections from areas 24 and 32. Specifically, the following steps should be executed to clarify the issue:

1) Perform the mixing analysis and generate "predicted site 4" connectivity from areas 24 and 32.

2) Calculate ALL formal hubness metrics for this "predicted site 4 profile", namely, degree and entropy.

3) Compare how well they fit to the actual site 4 metrics.

4) If the "predicted site 4" hub metrics fit better to the actual site 4 than the controls of e.g., mixing area 24 with randomly selected areas (9/46v, 45, 44….etc. that is, all 27 areas apart from areas 24 and 32) then the we have some evidence that the hubness DOES occur due to mixing, even if there is no pure, at least linear, addition of connectional profiles.

Note, that KL-divergence between predicted site 4 and actual site 4 connectivity might be low but degree centrality similarity might be even higher (the same with entropy of connections), since all these metrics do not quantify the same aspects.

Importantly, all the above controls cannot exclude that within area inhomogeneities lead to the observed hubness (stated in the previous round of revisions). This is impossible in my view, since we cannot know the structure and spatial periodicity of projection neurons within these areas across animals and how they are adding up from case to case when injections involve two areas and if they lead to what we observe from cases from different animals.

I think that it is fair to run the controls that are mentioned above and, depending on the outcome, support site 4 as hub but also indicate the methodological limitations in the manuscript (site 4 might be a hub, but…). It is more than fair to do so – invasive-tract tracing is the gold standard, but like any method, it has its limitations and we should operate with such limitations.

I hope that the above comments are of help.

---

## [Author Response]

Essential revisions:The reviewers saw considerable value in this study. If verified, the gradient of connectivity and "hubness" of rACC would be an important and novel finding. Furthermore, there was an appreciation of the use of convergent methods between high resolution tracing and diffusion tractography for cross-species comparison.However, the reviewers were also in agreement about the primary weakness of the study: The number of injections was highly limited and so this has left open several alternatives to a hub in rACC. For example, beyond some basic methodological concerns, there could be sites in the caudal regions left untested, such as dorsal ones, that would show a similar hub-like pattern, or there could be more than one region being conflated in what is being called rACC (site 4). Thus, the overall consensus was that more injections are needed in order to draw the broad conclusion that rACC is a connectional hub. Here are specific concerns along these lines from reviewers:1) Three different tracers are used in the study, but it was not indicated which was injected where. Could the findings of strong labeling in more areas after a single injection in site 4 be attributed to a more efficient tracer, better uptake at the injection site, or a stronger antibody used in the processing of the tissue in each case? With one case per site and missing methodological detail, it is not possible to accept such a broad conclusion.

The tracers injected into sites 1–6 are: FR, LY, FS, FR, LY and FR, respectively. The experimental design took into account the variability of tracer efficiency, uptake level and antibody reactivity, described below in detail. The findings of extensive labeling of an injection into site 4 could not be attributed to these factors.

First, the type of tracers is randomized among cases. Sites 1, 4 and 6 were injected with FR. In contrast to site 4, site 1 showed the most limited labeling among all six cases, with the fewest areas contributing to 50% of its inputs (Figures 3 and 4). Thus, the tracer FR is not more or less efficient than LY or FS. Moreover, this is consistent with our experience with these tracers. Our entire database of injection sites in non-human primates includes 54 cortical injections of LY, 34 of FR, and 29 of FS, all with outstanding transport. We find similar transport results regardless of the tracer used. We have included the tracer information for each site in the revised Results “Overview*”*.

Second, although uptake and transport are difficult to quantify, the extensive labeling of case 4 is unlikely to be attributed to these factors for the following reasons: 1) All tracers were conjugated to dextran amine (provided by Invitrogen) to optimize the transport and stability (subsection “Anatomical tracing experiments”, first paragraph) (Rajakumar et al., 1993). 2) Prior to selecting a case for analysis, we first assessed its transport to their known and distant connections, including the posterior cingulate/retrosplenial cortex and the thalamus. All cases showed dense labeling in areas 23, 29 and 30, and in the AV, VA and MD nucleus in the thalamus. We have included additional details of the qualitative assessment in the revised Materials and methods and Results (subsection “Analysis: strength of inputs and defining the hub”, first paragraph; Results subsection “Overview”). 3) If the uptake level was particularly high in a case, it would result in a greater large number of labeled cells in that case. We did not observe that for case 4 compared with the other cases (Figure 7—figure supplement 1B). 4) Importantly, if that had occurred, it would not affect the relative numerosity of labeled cells across areas. The input strength to each site is measured by the percentage of labeled cells to take into account possible variability of uptake.

2) The conclusion that only one site in the cingulate cortex is a connectional hub is based on injection sites located mainly in the ventral bank of the cingulate sulcus (sites 5 and 6), whereas the site in rACC covers both the ventral and dorsal banks. Although it has been suggested that the fundus of the cingulate sulcus might be the dorsal limit of the cingulate cortex (Vogt et al., 2005), this dorsal limit has been challenged. There is functional and anatomical evidence that the dorsal bank of the cingulate sulcus belongs to the cingulate cortex (see Procyk et al., 2016 for review). For example, all electrophysiological studies demonstrating activity related to cognitive control and performance monitoring have recorded within the dorsal bank of the cingulate sulcus in monkey. So, can the authors exclude that another hub may exist in the dorsal bank of the cingulate sulcus (posteriorly to site 4, for example at the level of sites 5 and 6)?

We agree that there may be more than one hub within the ACC region. It was not our intention to conclude that site 4 is the only hub in the ACC. As suggested by the reviewers, we examined an injection site (control case 5b) in the dorsal bank of the cingulate sulcus at the level between sites 5 and 6. The degree-centrality measured by the number of regions was similar between site 5b and sites 1, 2 and 6. The limited degree-centrality of site 5b did not support the hypothesis that this site in the dorsal bank is a hub. The location and photo micrograph of site 5b have been added to Figure 1A and B. The stereology results have been included in the revised Results, “Projection patterns of the control cases”. Figure 7 has been updated accordingly.

We have included additional discussion of this new result and the possibility of other hub sites in the ACC:

“Potential hubs in other subregions of the rACC. The results in this study did not suggest that only one hub exists in the rACC. […] This heterogeneity is consistent with findings in our other cases and in other cytoarchitectonic areas (Luppino and Rizzolatti, 2000), where inputs are derived from more than one functional region.”

3) Site 4 was at the intersection of areas 24 and 32 and sites 1-3 and 5-6 are located in areas 32 and 24, respectively (Results, first paragraph). Is it relevant that the injection in site 4 involves 2 areas and thus appears as a hub? The control case is intended to demonstrate that it is the location and not the size of the injection that matters, but the control case involves more posterior areas with less diverse profiles and number of connections. The fact that site 4 involves both areas 32 and 24 possess a problem since its "hubness" might be the result of the convolution of the connectivity profile of the two sites (3 and 5) with a high number of connections and diversity of connectivity profiles as Figure 4 indicates. The arguments against such an interpretation (subsection “Input patterns vs. cytoarchitectonic patterns”) are unconvincing. The first one does not provide evidence against the aforementioned interpretation. The second one assumes a simple sum of the connectivity profiles of two areas. But, this assumption is not necessary since there is connectional heterogeneity within areas (see for instance Luppino et al. work for the frontal cortex). In other words, there is not pure addition of connectivity profiles.

This is an interesting point. Indeed, several studies have demonstrated functional and connectional heterogeneity within cytoarchitectonic areas (e.g. Luppino and Rizzolatti, 2000). Our results from case 5b (dorsal bank site) showed such heterogeneity in that it received inputs from both vlPFC and premotor areas. However, its degree-centrality did not suggest a hub. To address the issue more robustly, we performed additional analyses to examine: (1) whether the inputs to site 4 can be accounted for by a composition of the connectivity profiles of the cytoarchitectonic areas 32 and 24, and (2) whether the spatial pattern of inputs to site 4 is a composition of those of its neighboring sites 3 and 5. The results suggested “no” to both scenarios.

If the projection pattern changes gradually from areas 32 to 24, then the strength of the major FC projections to site 4 would fall between the strength of these FC projections to areas 32 and 24. Moreover, if the input pattern of site 4 was a composition of inputs to areas 32 and 24 but one area contributed more than the other, then this inequality should be present consistently across FC areas. In other words, one would expect to see a closer approximation of projection strength between site 4 and either area 32 or 24 but consistently across all FC areas.

To test these predictions, we compared the strength of projections to site 4 with the average strength of projections to area 32 (averaged percent scores of sites 1–3) and those to area 24 (averaged percent scores of sites 5 and 6). We found that 8 areas, 14O, 11, 47L, 9/46D, 9L, 9M, 9/32 and 8A showed stronger projections to site 4 than to either area 32 or 24 (updated Figure 8A). Moreover, 14 out of the 27 FC areas showed more similar projection strength between site 4 and area 24 than site 4 and area 32 (Figure 8A). The 14/27 ratio indicates that the similarity between site 4 and area 32 or 24 was not consistent across FC areas but a random pattern. These observations do not support the above predictions of the “composition” hypothesis.

We further compared the projection pattern of site 4 with different weighted averages of the connection patterns of areas 32 and 24. We used the symmetric Kullback-Leibler distance as a measure of similarity between the normalized connectivity patterns. The best fit to site 4 was a combination of approximately 2/3 of area 24 and 1/3 area 32. However, the distance measures were indistinguishable from a situation where we have replaced one of the two areas’ connectivity patterns with a uniform distribution over the FC areas (Figure 8B). In addition, the optimum mixture was inconsistent across different FC areas, and in many samples of different weights the connectivity of site 4 was higher/lower than predicted by the weighted average model (Figure 8C). This inconsistency shows that site 4 connectivity is not a simple linear mixture of the input patterns of areas 32 and 24.

In the second analysis, we compared the spatial patterns of labeled cells across sites 3, 4 and 5. In the reconstructed 3D model of the retrogradely labeled cells (Figure 8D), site 4 showed a high number of cells in the caudal dlPFC and the principal sulcus, which were only sparsely covered by cells projecting to the neighboring sites 3 and 5. This result argues against the alternative interpretation of the "hubness" of site 4, that it is a result of the convolution of the connectivity profile of the two neighboring sites.

Details of this analysis are included in the updated Materials and methods:

“Comparison of the inputs to site 4 with a composition of inputs to areas 32 and 24. Based on the analyses described above, site 4 stands out as having a uniquely diverse input pattern as compared to the other sites[…] *P_s4_(a*) is the probability composed of *P_32_* and *P_24_* with weight *w* (0 ≤ *w* ≤ 1).”

The method for 3D reconstruction is described as follows:

“For each case, the stack of 2D coronal charts from StereoInvestigator were imported into IMOD, a 3D rendering program (Boulder laboratory for 3D Electron Microscopy). […] The 3D model of each case was then merged through spatial alignment to our in-house template brain as previously described (Haber et al., 2006).”

The results described above are added to the Results section. Discussion of these results is added as follows:

“A hub or inputs reflecting the junction of two cytoarchitectural areas?An alternative to the hub hypothesis is that site 4 was at the border of areas 32 and 24, and thus, its diverse inputs could be a result of a composition of inputs to areas 32 and 24. […] This cannot be predicted by the location of this site among the cytoarchitectonic areas.”

[Editors' note: further revisions were requested prior to acceptance, as described below.]

The manuscript has been improved but there are some remaining issues that need to be addressed before acceptance, as outlined below:One of the main points raised previously was the claim that the "hubness" of site 4 could not reflect an injection involving 2 areas, that is, area 24 and 32. Though the control provided rules out an additive version of this account, there may be reason to doubt additivity of connectivity. Thus, reviewer 3 has suggested a set of additional control analyses to fully rule out this concern. These are appended in this reviewer's comments below. Reviewer 1 also has a suggestion regarding the strength of the conclusions that you should consider in your revision. In discussion, the reviewers were in agreement that both of these points should be addressed. Thus, I send this back to you for additional revision. The reviewers' specific comments follow.Reviewer #1:The authors addressed key points raised and concluded that due to technical limitations, there indeed could be other areas within the anterior cingulate region with strong connectivity resembling a 'hub'. In view of this conclusion, it would be consistent to also tone down the initial part of the Discussion by removing words such as 'strongly' as in: "..the integrative nature of site 4 strongly supports that it is a hub in the frontal cortex..".The manuscript varies in style of expression. For example, the first paragraph of Discussion reads like a section from Results. There is variation in the style of the references in the text, missing references from the reference list and misspellings of some names.Reviewer #3:I would like to thank the authors for their clarifications.I would like to focus on one of the main points, namely, the claim of site 4 being a hub and that this is not explained by the fact that the injection involves 2 areas, that is, area 24 and 32.The authors performed a nice analysis showing that site 4 connectivity cannot be explained by a linear addition of the pattern of connections of areas 24 and 32.This is a basic demonstration of the fact that the assumption of "pure addition" of connectivity profiles does not necessarily hold. This was stated in my previous comment: "But, this assumption is not necessarily true since there is connectional heterogeneity within areas (see for instance Luppino et al. work for the frontal cortex). In other words, there is not pure addition of connectivity profiles."So the authors indeed show that there is no pure linear addition as I indicated, but this is something that is distinct from claiming that the overlap might not contribute to the "overshoot" in the metrics that are used to classify sites as hubs.A more direct rigid control analysis addressing the hubness of sites should focus on the criteria used for hubness and if they could be synthesized and to what extend they resemble site 4 hubness by mixing connections from areas 24 and 32. Specifically, the following steps should be executed to clarify the issue:1) Perform the mixing analysis and generate "predicted site 4" connectivity from areas 24 and 32.2) Calculate ALL formal hubness metrics for this "predicted site 4 profile", namely, degree and entropy.3) Compare how well they fit to the actual site 4 metrics.4) If the "predicted site 4" hub metrics fit better to the actual site 4 than the controls of e.g., mixing area 24 with randomly selected areas (9/46v, 45, 44….etc. that is, all 27 areas apart from areas 24 and 32) then the we have some evidence that the hubness DOES occur due to mixing, even if there is no pure, at least linear, addition of connectional profiles.Note, that KL-divergence between predicted site 4 and actual site 4 connectivity might be low but degree centrality similarity might be even higher (the same with entropy of connections), since all these metrics do not quantify the same aspects.Importantly, all the above controls cannot exclude that within area inhomogeneities lead to the observed hubness (stated in the previous round of revisions). This is impossible in my view, since we cannot know the structure and spatial periodicity of projection neurons within these areas across animals and how they are adding up from case to case when injections involve two areas and if they lead to what we observe from cases from different animals.I think that it is fair to run the controls that are mentioned above and, depending on the outcome, support site 4 as hub but also indicate the methodological limitations in the manuscript (site 4 might be a hub, but…). It is more than fair to do so – invasive-tract tracing is the gold standard, but like any method, it has its limitations and we should operate with such limitations.I hope that the above comments are of help.

Regarding further control tests for “hubness”, i.e. whether the patterns of site 4 can be predicted by non-linear mixing of patterns of areas 32 and 24, we performed additional analyses following the steps suggested by reviewer 3. Below is the original comment quoted in italic followed by our response.

*1) Perform the mixing analysis and generate "predicted site 4" connectivity from areas 24 and 32*.

We generated 1000 samples of “predicted site 4” connectivity by mixing the connectivity profiles of areas 24 and 32. The mixing used varying weights across frontal cortical (FC) areas. Thus, the mixing resulted in non-linear combinations rather than a pure addition of the profiles of areas 24 and 32. The mixing procedure is included in the revised Materials and methods (subsection “Comparison of the inputs to site 4 with the composition of inputs to areas 32 and 24”.

2) Calculate ALL formal hubness metrics for this "predicted site 4 profile", namely, degree and entropy.

We calculated the number of FC areas that contributed to 50% and 75% of the total input (degree-centrality at 50% and 75%, respectively) and entropy for each of the 1000 samples.

3) Compare how well they fit to the actual site 4 metrics.

Site 4 has 5 FC areas contributing 50% inputs, and 10 FC areas contributing 75% inputs (Figure 4). Among the 1000 generated samples, only 38 showed 5 or more FC areas contributing 50% inputs, and 63 showed 10 or more FC areas contributing 75% inputs. Moreover, 117 of the 1000 samples showed greater entropy than that of site 4.

4) If the "predicted site 4" hub metrics fit better to the actual site 4 than the controls of e.g., mixing area 24 with randomly selected areas (9/46v, 45, 44….etc. that is, all 27 areas apart from areas 24 and 32) then the we have some evidence that the hubness DOES occur due to mixing, even if there is no pure, at least linear, addition of connectional profiles.

We mixed the connectivity pattern of area 24 or 32 with a uniform distribution, generating 1000 control samples in each case. In both control cases, more than 600 samples showed degree-centrality ≥ 5 at 50%, and more than 900 showed degree-centrality ≥ 10 at 75%. More than 900 samples showed greater entropy than that of site 4. Thus, the degree and entropy of site 4 were better predicted by the mixture with a uniform distribution than the combination of areas 32 and 24. Taken together, these results suggest that the hub properties of site 4 did not occur due to the mixing of connectivity patterns of areas 32 and 24.

The above results are added to the subsection “Comparison of the inputs to site 4 with the composition of inputs to areas 32 and 24”. The Discussion is updated accordingly.

We have also followed the suggestions by reviewers 1 and 3 to adjust the conclusion and state the methodological limitations. The conclusion is changed to “…the integrative nature of site 4 suggests that it is a hub in the frontal cortex.” In addition, a new subsection is added to the Discussion:

“Methodological limitations.A main limitation of the current study is that experiments were performed in different animals. […] Thus, when observed as a whole, the region containing these neurons may appear as a hub.”

Finally, following the suggestion by reviewer 1, we have revised the style of expression in the Discussion. The first two paragraphs are now merged into one in the revised version. The references are also reformatted to be consistent throughout the text.